

# Aircraft surveys for air eDNA: probing biodiversity in the sky

Kimberly L. Métris[1,2] and Jérémy Métris[2]

[1] Department of Genetics and Biochemistry, Clemson University, Clemson, SC, United States
[2] Airborne Science LLC, Clemson, SC, United States

## ABSTRACT

Air is a medium for dispersal of environmental DNA (eDNA) carried in bioaerosols, yet the atmosphere is mostly unexplored as a source of genetic material encompassing all domains of life. In this study, we designed and deployed a robust, sterilizable hardware system for airborne nucleic acid capture featuring active filtration of a quantifiable, controllable volume of air and a high-integrity chamber to protect the sample from loss or contamination. We used our hardware system on an aircraft across multiple height transects over major aerosolization sources to collect air eDNA, coupled with high-throughput amplicon sequencing using multiple DNA metabarcoding markers targeting bacteria, plants, and vertebrates to test the hypothesis of large-scale genetic presence of these bioaerosols throughout the planetary boundary layer in the lower troposphere. Here, we demonstrate that the multi-taxa DNA assemblages inventoried up to 2,500 m using our airplane-mounted hardware system are reflective of major aerosolization sources in the survey area and show previously unreported airborne species detections (*i.e.*, *Allium sativum* L). We also pioneer an aerial survey flight grid standardized for atmospheric sampling of genetic material and aeroallergens using a light aircraft and limited resources. Our results show that air eDNA from terrestrial bacteria, plants, and vertebrates is detectable up to high altitude using our airborne air sampler and demonstrate the usefulness of light aircraft in monitoring campaigns. However, our work also underscores the need for improved marker choices and reference databases for species in the air column, particularly eukaryotes. Taken together, our findings reveal strong connectivity or mixing of terrestrial-associated eDNA from ground level aerosolization sources and the atmosphere, and we recommend that parameters and indices considering lifting action, atmospheric instability, and potential for convection be incorporated in future surveys for air eDNA. Overall, this work establishes a foundation for light aircraft campaigns to comprehensively and economically inventory bioaerosol emissions and impacts at scale, enabling transformative future opportunities in airborne DNA technology.

Corresponding author
Kimberly L. Métris,
kimberlymetris@gmail.com

## INTRODUCTION

A rapidly expanding noninvasive technique for monitoring taxa of conservation, health, or agricultural importance is sampling DNA released from organisms into the surrounding environment, such as water and glacier, sediment, precipitation, smoke, or air (*Ficetola et al., 2008*; *Varotto et al., 2021*; *Palacios Mejia et al., 2021*; *Ladin et al., 2021*; *Kobziar et al., 2019*; *Serrao et al., 2021*). Air carries nucleic acid-based particulate matter (bioaerosols). Bacteria and viruses, plant parts including pollen and seeds, fungal spores, invertebrates, vertebrates, and extra-organismal fragments of cells, tissue, feces, or other excretions—all with detectable signatures from their genetic composition—move through the atmosphere. Accordingly, the ability to survey the airborne metagenome representing all taxa, as well as targeted detection within this background, would offer wide applications for pest, infectious agent, and aeroallergen detection, wildlife, ecosystem and climate monitoring, industrial hygiene assessment, and bioprospecting of culturable organisms for agricultural improvement or bioremediation (*Bohmann et al., 2014*; *Tanaka et al., 2019*).

Aerobiology is an established research discipline focused on aerosolized bacteria as well as pollen and fungi as aeroallergens or pathogens (*e.g.*, *Sánchez-Parra et al., 2021*; *Rowney et al., 2021*; *Banchi et al., 2020a*; *Kraaijeveld et al., 2015*; *Longhi et al., 2009*; *Abrego et al., 2018*; *Tordoni et al., 2021*) including crop pathogen monitoring (reviewed in *Mahaffee & Stoll (2016)*). Important biological associations on Earth's surface may be maintained in the air. For example, bacteria and their endotoxins can be co-transported with pollen or fungi in air currents and exacerbate the immune response (*Oteros et al., 2019*), suggesting co-dispersal is an important ecological phenomenon in the aerobiome (*Morris et al., 2007*). When used as untreated fertilizer, livestock manure can be a source of infection and zoonosis, containing bacteria, protozoa, and viruses responsible for human or animal disease outbreaks (*Polley et al., 2022*). Bioaerosol emissions from wastewater treatment plants (*Pascual et al., 2003*) and within hospitals (*Pertegal et al., 2023*) can contain airborne pathogens. Scientists have used airborne bacteria as indicators for surface emissions of aerosolized human wastewater and cow manure (*Bowers et al., 2013*). Animal waste and particulate matter themselves are aeroallergens which can induce immune response (*Zahradnik & Raulf, 2017*). Yet no aerobiome studies have investigated the association of rural-agroecosystems and suburban areas using metabarcoding to identify key sources (livestock/poultry and plants) in conjunction with bacterial taxa, and little is known about connectivity among air eDNA assemblages. Such advances in comprehensive biomonitoring would enable us to mitigate aeroallergen impacts on human health (*Rowney et al., 2021*) and better understand Earth's aerobiome.

Despite the fundamental role of air at both local and global scales, the atmosphere is a mostly unexplored medium for environmental nucleic acids (eDNA or eRNA) and the full biodiversity of the aerobiome remains understudied. Recently a few groups have used air for environmental sampling of animal DNA in captive housing enclosures and zoos (*Clare et al., 2021*, *2022*; *Lynggaard et al., 2022*; *Serrao et al., 2021*). These proof-of-concept studies demonstrated that it is possible to collect and sequence airborne DNA from animals in artificially confined environments. Airborne eDNA capture in controlled

conditions offers clearly interpretable results but such enclosures may not fully represent the atmosphere at large. Further, most studies of air DNA from vertebrates and plants have been restricted to collections at or near ground level, often from a stationary location (*Clare et al., 2021*, *2022*; *Lynggaard et al., 2022*; *Serrao et al., 2021*; *Johnson, Cox & Barnes, 2019a*, *2019b*; *Klepke et al., 2022*; *Johnson et al., 2023*), although one study sequenced plant DNA up to 1,000 m above ground level (AGL) (*Sánchez-Parra et al., 2021*). Studies of airborne dust or eDNA from animals at ground level have reported that distance to the sampling device and high biomass increase probability of detection (*Lynggaard et al., 2022*), yet it is unknown whether any vertebrate DNA is detectable thousands of meters into the atmosphere.

Monitoring bioaerosols starts with their direct and scalable capture from the atmosphere, including higher altitudes and less accessible places. Several creative approaches have been taken to retrieve bioaerosols from the air using aircraft. One group used impaction sampling to estimate the abundance of pollen and fungi up to 2,000 m by holding at arm's length adhesive-coated microscope slides out of the window of a light aircraft in Greece, but genetic analyses were beyond the scope of the study (*Damialis et al., 2017*).

Genetic-based analyses of airborne nucleic acids in bioaerosols require purpose-built air samplers. For example, a high flow impaction sampler with Petri dishes coated in petroleum jelly was used to assess vertical distribution of plants, bacteria, and fungi up to 1,000 m in Spain from a light aircraft with high-throughput DNA sequencing (*Sánchez-Parra et al., 2021*). *Sánchez-Parra et al. (2021)* adapted the commercially available DUO SAS Super 360 impaction air sampler to be mounted on an aircraft wing strut with its head inlet holes facing backward, which they stated was to avoid air turbulence from the propeller and cross-contamination. While there was a delayed start option, the collection surface was always exposed to the environment.

Researchers in Spain outfitted a CASA-212 turboprop with an air intake on the roof of the aircraft and a filter holder, flow meter, and vacuum pump inside the cabin (*Aguilera et al., 2018*; *González-Toril et al., 2020*). The filter holder inside the cabin was designed to be replaced in flight, which could potentially allow for contamination of the sample. Researchers also collected bioaerosols for pyrosequencing outside a DC-8 jet airliner by actively filtering air from a pitot-style inlet on the fuselage (*McNaughton et al., 2007*) through a cellulose nitrate membrane in the cabin using a vacuum pump with a rated flow of 20 L·min$^{-1}$ (*DeLeon-Rodriguez et al., 2013a*). In this system, contamination from the inlet lines and/or aircraft cabin was possible (*DeLeon-Rodriguez et al., 2013b*).

*Smith et al. (2018)* presented a cascade sampler called the Aircraft Bioaerosol Collector (ABC) installed on a C-20A (Gulfstream III) jet to collect microorganisms at altitudes up to 30,000 ft in the western US. Their sampler uses ram air (not a vacuum pump) to pass air from a pitot-style inlet on the outside of the aircraft fuselage through gelatinous membranes on stages in a cascade system inside the aircraft cabin, with a mean reported flow of 8.5 L·min$^{-1}$ and includes a ball valve to prevent air flow during takeoff and landing. This study reported limitations to the data given the possible influence of contaminants from their collection hardware, as there was no way of testing whether bioaerosols

remained in inlet lines as carryover prior to sampling and influenced their results (*Smith et al., 2018*).

Aerobiology research campaigns are conducted typically from jets and other heavy aircraft (*Smith et al., 2018*; *Jaing et al., 2020*; *DeLeon-Rodriguez et al., 2013a*) that are costly to operate; few bioaerosol studies have used light aircraft as a tool (but see *Trägårdh (1977)* and *Damialis et al. (2017)*), and we could identify only one that used next-generation sequencing (*Sánchez-Parra et al., 2021*). Studies of biological inocula in the atmosphere face several existing challenges—often due to design flaws—leading to fragmented or damaged DNA, premature dissolving of gelatinous filters, contamination of the sample pre- or post-sampling with foreign DNA not part of the air sample, and lack of real-time flow measurements, efficiency, and scalability (discussed in *Després et al. (2012)* and *Smith et al. (2018)*).

Bioaerosol sampling achievements, prospects, and challenges are reviewed elsewhere (*Mainelis, 2020*; *Šantl-Temkiv et al., 2020*), but the primary goal of air samplers is to extract a representative sample and preserve its properties for analysis. Sampler operators also should take steps to minimize particle loss during sampling since they affect accuracy (*Mainelis, 2020*). Atmospheric capture of nucleic acids thus far shares similar hurdles to the aquatic realm, including potentially low biomass in an "ocean of air" as well as contamination prevention and assessment, but other challenges also exist for aircraft platforms used in genetic research. For example, achieving isokinetic conditions desired for sampling can be difficult for high velocity aircraft (*Aguilera et al., 2018*). Contamination is a persistent challenge in aerobiology research (*Smith, 2013*). The sampler must be installed on the airplane in a location that avoids chemical contamination from flight operation and is protected from foreign genetic material or cross-contamination (*i.e.*, not from the air being sampled or insects clogging the system, or non-biological particles). Air intakes that are on the fuselage of the aircraft are difficult to sterilize (*Aguilera et al., 2018*). Ease of cleaning or disinfecting the sampler between runs should be considered to minimize sample contamination (*Mainelis, 2020*). Sampler operability should be tested in diverse field environments that represent the actual environments where the sampler will be used (*Mainelis, 2020*). Indeed, while sequencing of bioaerosols might offer a powerful tool for simultaneous surveys of terrestrial biodiversity across lifeforms, it will be imperative to carefully control for possible contaminants (*Bohmann & Lynggaard, 2023*). These challenges must be overcome for the scientific community to access transformative opportunities surrounding airborne DNA collection directly from the atmosphere efficiently at scale.

Our aim was to design a robust, sterilizable hardware system for airborne nucleic acid capture featuring active full-flow filtration of a quantifiable, controllable volume of air and a high-integrity chamber to protect the sample from loss or contamination. We used our hardware system to sample multiple height transects over major aerosolization sources by aircraft and filter representative bioaerosols directly from the atmosphere (not as passive collection of accumulated dust deposition over time), coupled with amplicon sequencing of a combination of multiple DNA metabarcoding markers targeting bacteria, plants, and vertebrates, to demonstrate the operability of our system and to test the hypothesis of their

large-scale presence and detectability thousands of meters into the planetary boundary layer, PBL.

To examine our hypothesis, high throughput amplicon sequencing (HTAS) was performed using an Illumina® MiSeq platform (2 × 250 bp) on bacterial, plant, and vertebrate DNA isolated simultaneously from atmospheric air samples from duplicate filters captured on dedicated research flights up to 2,500 m.

## MATERIALS AND METHODS

### Sterile high integrity capture probe development

We developed a high integrity capture probe and its supporting system to sample genetic material directly from free stream atmospheric air (Fig. 1). This sampling system features three sub-assemblies: a mast with static port inlets, intake valve, filter isolation chamber and discharge valve (hereafter the "probe"); flow meter, vacuum gauge, vacuum release valve and muffler (hereafter the "manifold"); and vacuum pump and electric motor (hereafter the "vacuum pump"). All air sampled by the probe must pass through the filter which in this case was used to capture nucleic acids. We designed the probe to safeguard initial sterility of the filter pre-sampling and shield the collected genetic material post sampling. The probe is controlled by the operator using the manifold and vacuum pump located inside the aircraft cabin providing precise control over the exact location and time of sampling. Sampling is initiated by turning on the vacuum pump and closing the vacuum release valve, thus allowing atmospheric air through the probe where bioaerosols are captured on the membrane filter (Fig. 1). The air then flows into the manifold where pressure and flow are monitored and exits the vacuum pump air discharge back to the atmosphere. Sampling is stopped by turning off the vacuum pump and opening the vacuum release valve. The filter isolation chamber accepts standard 47 mm collection media for compatibility with next-generation sequencing, microscopy, and many culture-based techniques. In this study, the probe is mounted outside the aircraft attached to the wing strut (Fig. 2).

This system offers improvement over existing devices in that it features:

1. Safe and economical air sampling using a light aircraft;
2. A sterilizable rugged stainless-steel probe with no inlet lines to be contaminated;
3. A filter isolation chamber safeguards the initial sterility and the collected sample with no opportunity for contaminants to enter the isolation chamber either through the inlet or discharge valve when not actively sampling;
4. All cleaning, sterilization and handling of the filters with the probe opened is performed in a controlled environment under a bio safety cabinet and never has to be opened in the field preventing handling contamination;
5. Flow and pressure manifolds to record air volumes processed (our modular install can include other measurement devices);
6. The ability to be used in a stationary or moving manner, from aircraft or other vehicles, manned or unmanned; and

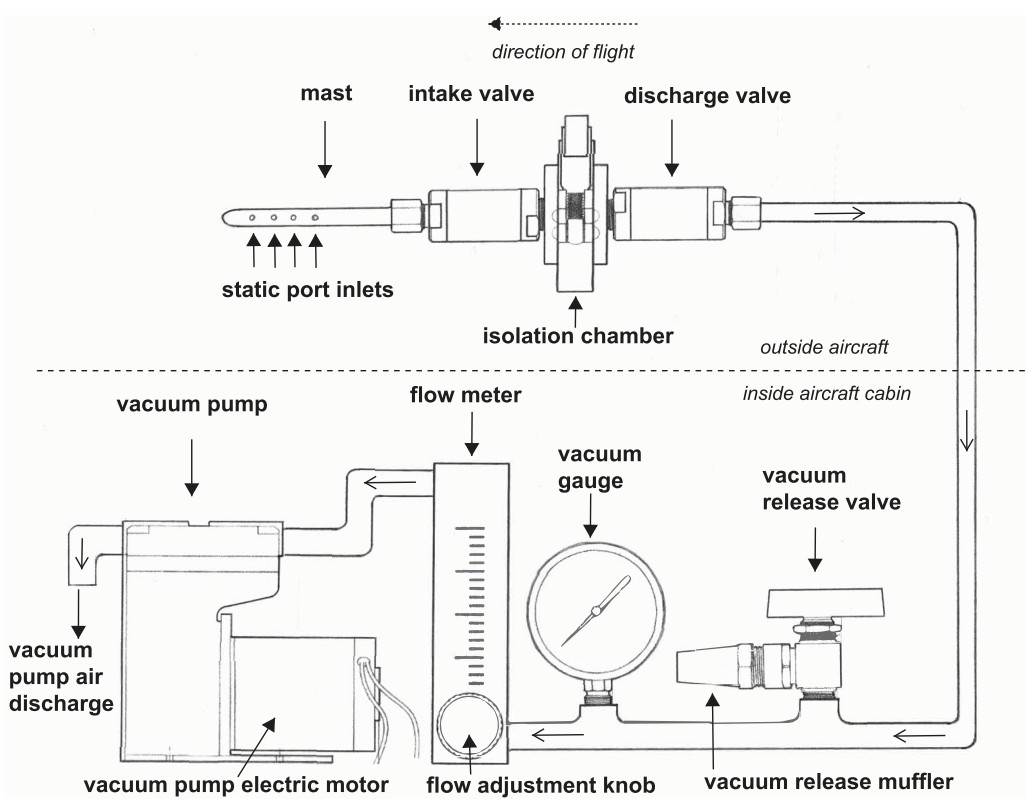

**Figure 1  A schematic drawing of our high-integrity probe and supporting system for air eDNA sampling.**                                 

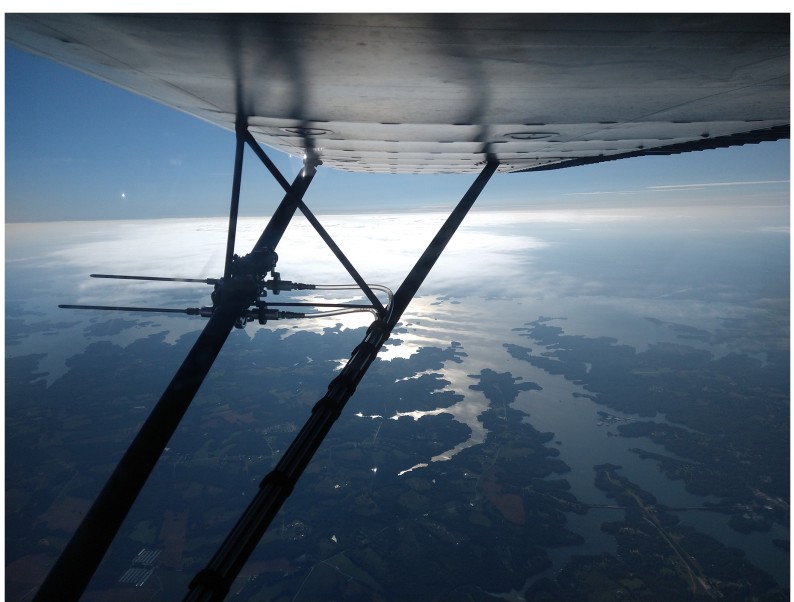

**Figure 2  Our portable high integrity sampler in action on a genetic aerial survey.** Image credit: the authors.                                 

7. The ability to accommodate various sample collection media and compatibility with various analysis methods depending on purpose and conditions.

In this study, we used a Maisi® 12V 3.5A diaphragm vacuum pump (model #GZ50-24) with -83 kPa vacuum and 33 L·min$^{-1}$ air flow. Air flow was measured using a Dwyer VFA-25SSV flow meter. Vacuum pressure was measured using an Ashcroft 733-52 pressure vacuum gauge. Air volume filtered per sample (L), and vacuum (-kPa) data were recorded for each sample in this study, including pre-survey test flights (Datas S1.1 and S1.2). During each research survey flight, we filtered 1,350–2,175 L of air with air flow rates typically between 10 and 15 L·min$^{-1}$ (Data S1.2) comparable to several successful aerobiology studies collecting microbes at ground level and from an observatory surface (~8.7 L·min$^{-1}$ in *Griffin et al., 2011*) and at 30,000 feet (8.5 L·min$^{-1}$) (*Smith et al., 2018*). To improve technical standards and comparisons across future studies, we encourage the scientific community to report actual measured flow rates, total air volumes, and the type of sampler used such as impaction or filtration as they have critically different operating airflow requirements.

We sourced specific hardware and materials to design a system that would be simple and durable that could withstand frequent outdoor usage as well as bleach sterilization, UV light and autoclaving to create a tool that could collect DNA using a light aircraft economically. We verified proper operation of the system at multiple points in our workflow, including confirming sampling chamber integrity and appropriate airflow. During cleaning before each equipment sterilization, we performed a leak test by filling both side of each probe intake and discharge valves with water and monitoring the change in water level over time. If any water drop was noted the valve would be discarded from use and replaced. We also vacuum leak checked the system periodically (the probe, manifold, and vacuum pump) by blocking the mast static inlet port, putting the system under max initial vacuum (−83 kPa) and noting the rise in pressure over time to find potential vacuum leaks which could degrade performance. Overall, this system represents a design tradeoff to maintain initial sterility, acceptable sampling flow rate, and guard the genetic material sampled from contamination and loss all while being safe and lightweight enough to be operated on light aircraft.

We designed the probe to be installable completely outside the aircraft cabin and fuselage away from any possible disturbance, such as propellers, engine emissions, or aircraft surfaces. Traditionally it has been common practice to change out membranes or filter cartridges in the cabin during aerobiological flights. However, we think this could potentially allow sample contamination (unless a portable hood is used inside the aircraft, but we cannot find any references to such in the literature to date). Thus, our customized probe was always sterilized and assembled in the lab under sterile conditions, and once installed outside the aircraft was never opened in flight. This may require sampling decisions to be made prior to the survey; from the authors' perspective as research pilots, this is a part of necessary flight planning.

In this study, the probe was mounted on the wing strut so that the forward-facing mast only samples free stream atmospheric air forward of the wing leading edge, uninfluenced

by aircraft surfaces (*King, 1984*) to prevent contamination of foreign DNA (apart from the sample) or chemicals from aircraft operation (Fig. 2). The probe longitudinal axis was installed parallel to the aircraft longitudinal axis for efficient collection in free stream air. All lines downstream of the probe discharge valve were routed along the strut to enter the fuselage through the right main landing gear leg and to the baggage area where the manifold and vacuum pump are located. Our choice of mounting location was determined by our type of aircraft and mission; the probe can be installed elsewhere.

To increase efficiency and reduce opportunities for contamination, ease of cleaning and disinfecting the sampler between runs is an essential requirement of our design. The probe is made of stainless steel and all components withstand common sterilization processes (*e.g.*, bleach, UV light, autoclaving). Sterilization of the device between collections (flights) after autoclaving and UV was performed by decontamination with 20% sodium hypochlorite (household bleach) in ultrapure water. Upon sterilization a new filter membrane was positioned, and the filter isolation chamber was closed rendering it ready for field use. After sampling, filters were extracted from the housing in a biological safety cabinet under maximum flow with sterile forceps decontaminated with autoclaving, 95% ethanol, and 20% bleach. Negative controls were included at the device sterilization and pre- and post-flight collection stages (see "Quality Assurance/Quality Control" section).

## Study area

The study area is situated in the southern foothills of the Appalachian Mountains in northwest South Carolina and northeast Georgia in the southeastern United States, representing 1,852 ha (4,576.39 acres) (Fig. 3). Georgia is ranked at the top for poultry production in the United States, with abundant livestock production adjacent and in surrounding states including South Carolina. This region is characterized by arable agricultural land (crop and food animal production), ecotone buffers, and some interspersed forests of pine stands and mature diverse mixed hardwoods in the canopy and subcanopy (*Cox & Straka, 2017*) as well as lakes, rivers, towns, and populated urban areas with hospitals and wastewater treatment plants. The study area was selected for its major aerosolization sources, including a multitude of poultry houses and livestock facilities, agricultural fields, industrial buildings such as wastewater treatment plants and hospitals, and urbanization or construction activity, with these ground-level emissions structures visible from the aircraft at all flight altitudes (Fig. 3; File S1). In addition, the region was selected for our familiarity with the terrain below our flight path as well as optimal efficiency to minimize time to initial sampling location/altitude and ensure legal, safe fuel reserves and alternate airport options.

## Aerial surveys

To develop an aerial survey technique specific to the purpose of genetic biosurveillance, we used the commercially available aviation application ForeFlight®. Prior to this study, we conducted test flights comprised of straightaways with minimal turns in relatively stable air to efficiently assess and optimize the flight path and duration for the research campaigns. The tracks for these test flights are provided in Data S1. A pre-study survey of

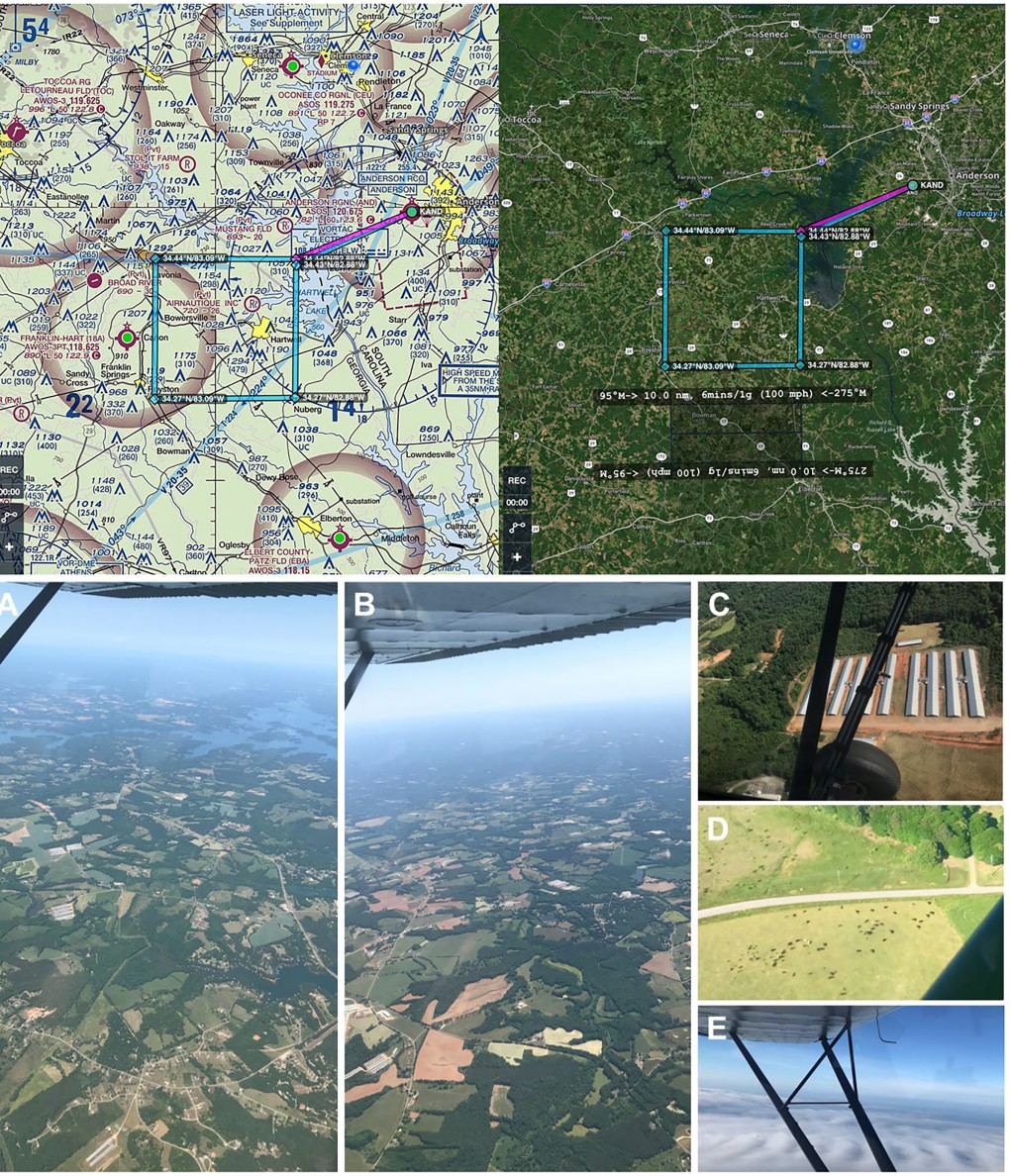

**Figure 3 Genetic aerial survey grid perimeter (top panel). Genetic aerial surveys in the southeast United States (bottom panel).** Top panel: The sides of each square measure 18.52 km (10 nautical miles). The area represents 18.52 km$^2$ or 1,852 hectares (4,576.39 acres). Map credit: © ForeFlight®. Bottom panel, clockwise from left: (A) view of study area from highest altitude sampled above the PBL at 2,500 m above ground level (8,500 ft mean sea level) and (B) from 1,200 m (4,500 ft MSL) with haze visible on horizon; view at 300 m (1,800 ft MSL) over (C) a poultry facility and (D) cattle on agricultural land; (E) view on a June survey flight with visible moisture as clouds. Photo credit: the authors.

the area at ground level (2 m) was also conducted (Data S1.4). The finalized aerial survey gridded transect we used to conduct all research flights is derived from the parallel racetrack pattern used over relatively flat terrain (*McConnell, Johnson & Burns, 2000*) and distances were selected to enable comprehensive yet efficient sampling in the chosen timeframe. The perimeter for our research flight surveys encompasses three-dimensional
space over the study area with specific ground emissions sources below our flight path, including plentiful poultry houses and cattle fields (Fig. 3). Flight tracks for all six research flights are provided in Fig. 4. We flew closely spaced gridded line transects 3–5 km (~2–3 nautical miles) apart with legs of 18.52 km$^2$ (10 nautical miles) flown in all four cardinal directions to limit the influence of wind vector direction on airborne eDNA capture (Fig. 4).

In May and June 2022, we flew aerial surveys to collect airborne DNA samples, departing from and returning to Anderson Regional Airport (Anderson, SC, USA). Filters were collected in duplicate at three altitudes, 1,800 ft mean sea level (MSL), 4,500 ft MSL, and 8,500 ft MSL (roughly 300, 1,200, and 2,500 m above ground level), with varied time of day for each altitude to minimize the effect of time of day in our study, which focuses on the influence of altitude and air mass history on multi-taxa bioaerosols (Fig. 4; Data S1.2). These altitudes were stratified throughout the PBL, which is approximately 1–2 km from the ground where the influence of Earth's surface is still detectable and mixing, diffusion, and transport of aerosols occurs (*Lenschow, 1986*; *Stull, 1988*; *Tegen et al., 2000*). The altitude of 2,500 m AGL or 8,500 ft MSL represents collection at high altitude (defined as at least 8,000 feet above sea level), consistently above the PBL and higher than other airborne genetic surveys of plant and vertebrate bioaerosols reported. Selection of these altitudes was also based on the aircraft's performance profile, best fuel economy, safety, and Federal Aviation Regulations. Surveys were flown at approximately 145–180 km h$^{-1}$ (90–110 mph) along the predetermined grid superimposed on the study area and presumed aerosolization sources (Fig. 3; File S1). Survey effort consisted of 7.5 h total flight time (1.25 h per flight × 3 altitudes per day).

## Flight telemetry data, atmospheric data, and airmass modelling

To determine the diversity and connectivity of airborne DNA from bacteria, plants, and vertebrates at different heights above Earth's surface in the open air using our sampling device, we conducted aerial surveys at three altitudes at or above the PBL. We consulted surface pressure analysis prognostic chart forecasts for each flight and recorded GPS position, altitude (MSL), aircraft ground speed (kts), wind velocity (kts) and direction (°), sky condition, air temperature and dewpoint (°C), and density altitude (ft) data during flight operations, in addition to airflow (L min$^{-1}$) (Data S1.2). Meteorological aerodrome (METAR) data from automated surface (weather) observation systems (ASOS) for all flights originating out of AND, Anderson Regional Airport (Anderson, SC, USA) and enroute near 18A, Franklin-Hart Airport (Canon, GA, USA) were recorded (Data S1.2). Our electronic flight bag consisted of the commercially available application ForeFlight® coupled with the Appareo Systems Stratus 3 (Fargo, ND, USA) portable Bluetooth wireless receiver used to obtain meteorological and telemetry data (weather and GPS information) and display Automatic Dependent Surveillance–Broadcast (ADS-B) data.

To determine whether our airborne DNA detections from bacteria, plants, and vertebrates in the atmosphere were influenced by flying through air masses with different transport histories (*i.e.*, how far and in what direction a parcel of air travelled in the time just prior to sampling the study region), we generated two-day kinematic back-trajectories

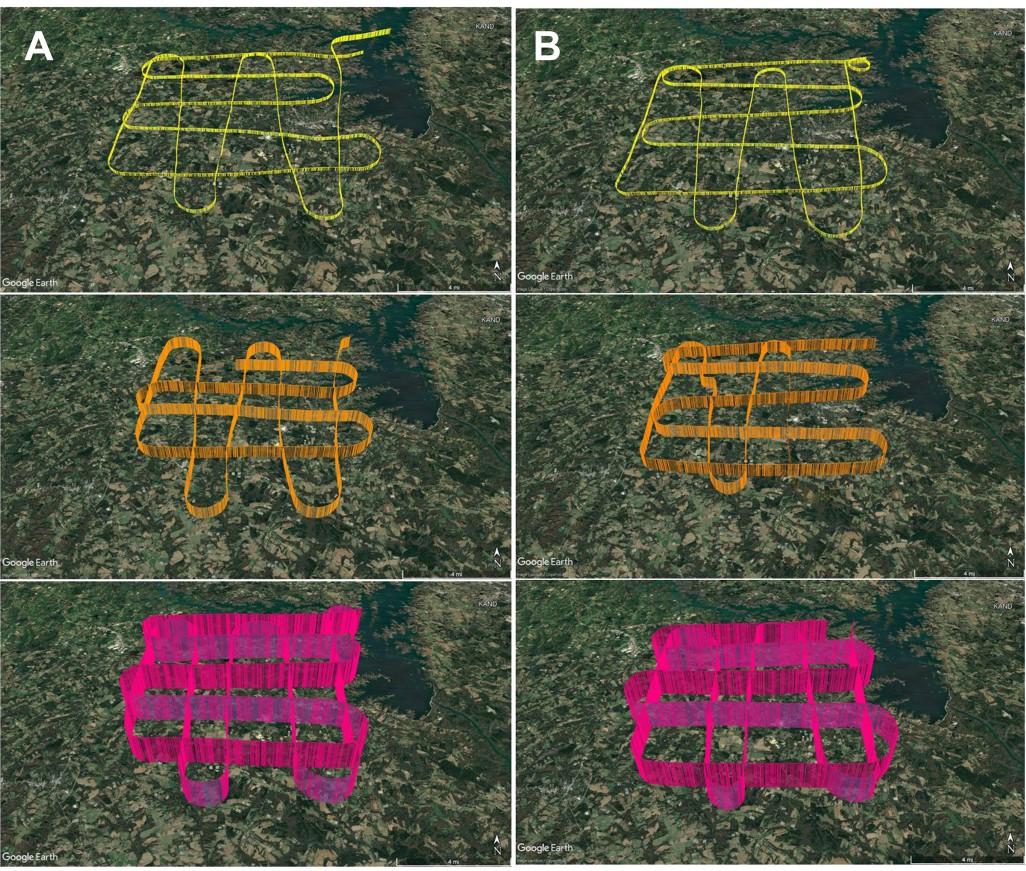

**Figure 4 GPS flight path tracks for genetic aerial surveys.** Flight path tracks for genetic surveys at 300 m (yellow), 1,200 m (orange), and 2,500 m (pink) on (A) 9 May and (B) 4 June. Flight patterns are based on GPS telemetry data recorded from ForeFlight®. Ten nautical mile legs were flown. Scale bar in lower right of each track represents four nautical miles. Test flight tracks in Data S1. Map credit: © Google Landsat/ Copernicus.

over our flight paths using the Hybrid Single-Particle Lagrangian Integrated Trajectory (HYSPLIT) model (*Stein et al., 2015*), which uses global meteorological data from the Global Data Assimilation System (GDAS) archive. Trajectories were run at three heights (300, 1,200, and 2,500 m AGL—corresponding to 1,800, 4,500, and 8,500 ft MSL). The HYSPLIT model is based on GDAS meteorological data at heights within the range of altitudes flown.

## Air filtration and genetic sample collection

For this study, our high integrity sampler was fitted with a sterilized cellulose nitrate membrane disc filter with a diameter of 47 mm and pore size of 0.22 μm (MDI Membrane Technologies, Inc., Harrisburg, PA, USA) housed in the isolation chamber (Fig. 1). Each sterilized sampling probe was used for one flight and altitude collection. Sampling was started once we reached the target and first GPS coordinate of the transect grid in straight-and-level flight by turning on the vacuum pump and closing the release valve (which opens the intake check valve) and terminated at precisely 75 min of sampling by simply reversing

the steps taken to begin sampling. To maximize the amount of air we sampled, we collected duplicate samples on each flight (filtering 1,350–2,175 L of air per sample) (Data S1.2).

## Quality assurance/quality control (QA/QC): comprehensive negative controls

We incorporated rigorous negative controls and exposed them to the same conditions encountered by our samples. These controls were obtained and processed from the microbiology lab during equipment *storage* (filters in each device with valves closed and pump off), equipment *cleaning* and *sterilization* (water used to flush the device after bleach rinse and from water left in an open 50 mL conical tube for 24 h in the hood prior to UV sterilization), from the *field* (filter in the device hardware with valves closed and pump off in ground operations and also in flight/airborne), and from the *molecular labs* during extraction and PCR (water left in an open 50 mL conical tube for 24 h in both the PCR room and the extraction room under the hood).

Negative and no template controls were processed in parallel with flight samples and included equipment storage, cleaning, and sterilization, airborne field collection, molecular work (*e.g.*, DNA extraction, PCR, sequencing library preparation), and bioinformatic analysis pipelines. Briefly, an extraction blank filter was processed with every batch of samples. Further, PCRs with sequencing adapters were performed with both negative controls (DNA extraction blanks) and no template controls (template replaced with nuclease free PCR grade water) for each primer set. Both negative controls for each primer set were purified and added to the sequencing pool even though no bands were visible on the gel. Also added to the negative control sequencing pool were the negative control samples from equipment sterilization and cleaning, from the field flights at altitude and in storage, and from the molecular labs as described above. For the most conservative approach to validate our instrumentation and data, we chose to discard all species for which we had a positive detection in the pooled negative control and drop them from analyses, instead of delimiting a detection threshold above which would be considered significant background contamination.

## DNA extraction

Equipment cleaning, filter removal, and DNA extraction were performed in a decontaminated biosafety cabinet with maximum laminar flow using sterile single-use nitrile gloves, a face mask, and protective clothing covering all exposed skin and hair. Equipment was autoclaved and rinsed with 95% ethanol and ultrapure water and sterilized using 20% bleach between samples. Filters were removed from the device using sterile forceps and stored at −80 °C until DNA extraction. DNA was extracted from duplicate filters for each of three flight altitudes on 2 days (12 samples total) using the E.Z.N.A.® Soil DNA Kit (Omega Bio-Tek, Norcross, GA, USA) following the manufacturer's protocol with the following modifications: after bead-beating, the filter was added to a homogenizer column (Omega Bio-Tek, Norcross, GA, USA) and the flow through was added back to the sample. DNA was eluted into 50 μL elution buffer, and five elution cycles were performed (with 5 min incubations at 65 °C) to increase nucleic acid concentration. We selected the

soil protocol because previous data from our lab (K. Métris, 2022, unpublished data) showed the soil protocol extracts DNA efficiently from our mélange of target taxa with improved yield compared to water, blood and tissue, plant, and CTAB methods. However, it impossible to know whether an extraction protocol we did not test would have performed better across all taxa. We recommend that future studies focused on one taxon use an extraction protocol for that specific target where possible to achieve optimal detection.

## High-throughput amplicon library preparation

Amplicon libraries were constructed using different universal primers for each taxon appended with 5′ Illumina® adapters: (F) 5′-ACACTCTTTCCCTACACGACGCTCTTCCGATCT-3′ and (R) 5′-GACTGGAGTTCAGACGTGTGCTCTTCCGATCT-3′: Bacteria: V341 (F): 5′-CCTACGGGNGGCWGCAG-3′; V805 (R): 5′-GACTACHVGGGTATCTAATCC-3′ (*Klindworth et al., 2013*), for the V3–V4 hypervariable region of the bacterial 16S ribosomal RNA gene; Plants: ITS2 (F): 5′-ATGCGATACTTGGTGTGAAT-3′, ITS4 (R): 5′-TCCTCCGCTTATTGATATGC-3′ (*White et al., 1990*; *Chen et al., 2010*; *Johnson, Cox & Barnes, 2019a*), for the plant-specific 5.8S—ITS2 gene region; Vertebrates: AquaF2 (F) 5′-ATCACRACCATCATYAAYATRAARCC-3′, Vr1d (R) 5′-TAGACTTCTGGGTGGCCRAARAAYCA-3′ (*Ivanova, Clare & Borisenko, 2012*) for partial amplification of the 5′ region of the mitochondrial cytochrome c oxidase subunit I gene. For two of the taxa listed, vertebrates and bacteria, we employed degenerate primers to amplify a broad range of templates. Our primer sets had been tested previously with different airborne DNA template concentrations quantified using a SpectraMax QuickDrop Micro-Volume Spectrophotometer (Molecular Devices, San Jose, CA, USA) and annealing cycles to ensure optimal amplification under the conditions used in this study. We used a one-step PCR approach with Illumina-tagged primers for vertebrate (*COI*) and plant (*ITS*) targets. Nested (two-step) PCR using 1 µL of untagged PCR product in a second PCR was performed only for 16S bacterial PCR products which were not visible on the gel otherwise.

Duplicate filter extracts were combined, and two PCR replicates were used per sample to reduce effects of PCR stochasticity and to optimize detection (*Alberdi et al., 2018*; *Polz & Cavanaugh, 1998*). For all taxa, each replicate PCR consisted of 12.5 µL GoTaq® master mix (ProMega Corporation, Madison, WI, USA), 7.5 µL PCR-grade $H_2O$, 3 µL of template DNA, and 1 µL of each tagged primer (10 mM stocks of each) in 25 µL final volume. Reaction conditions for the Vertebrate AquaF2/Vr1d primer set were as follows: 95 °C for 15 min, followed by 40 cycles of 94 °C for 30 s, 51 °C for 90 s, and 72 °C for 90 s and final extension at 72 °C for 10 min with a 10 °C indefinite hold. PCRs were run in duplicate with a positive control (*Ovis aries*, a species not known to occur at high biomass in our study area), as well as negative and no template controls. Reaction conditions for the Plant ITS2/four primer set were as follows: 95 °C for 10 min, followed by 40 cycles of 95 °C for 40 s, 49 °C for 40 s, and 72 °C for 40 s and final extension at 72 °C for 5 min with a 4 °C indefinite hold. PCRs were run in duplicate with positive (*Arabidopsis thaliana*), negative, and no template controls. Reaction conditions for the Bacterial 16S primer set were as follows: 95 °C for 10 min, followed by 25 cycles of 95 °C for 40 s, 55 °C for 2 min, and

72 °C for 1 min and final extension at 72 °C for 7 min with a 4 °C indefinite hold. PCRs were run in duplicate with positive (*Escherichia coli*), negative, and no template controls.

Combined PCR replicates were visualized along with positive, negative, and no template controls on 2% agarose gels with GelGreen® and 100 bp ladder. All negative controls appeared negative, and all positive controls showed expected amplicon sizes under a blue light transilluminator. Amplicons of expected length were excised with a sterile scalpel and purified using the E.Z.N.A.® Gel Extraction Kit (Omega Bio-Tek, Norcross, GA, USA). Pooled negative and no template controls were purified for sequencing using the column-based E.Z.N.A.® Cycle Pure Kit (Omega Bio-Tek, Norcross, GA, USA). We also performed Sanger sequencing verification of plant (*ITS +trnL*) and vertebrate (degenerate *COI+16S*) detections from the six research flights (described in Data S1.4).

### High-throughput sequencing on the illumina® Mi-seq platform

Multi-marker metabarcoding samples consisting of purified vertebrate *COI*, plant *ITS*, and bacterial 16S amplicon replicates representing each flight and altitude were quantified with a Qubit® fluorometer and pooled by flight in equimolar ratios normalized to a concentration of 20 ng/μL. Negative controls from equipment, flight, and lab (extraction blanks and no template PCR) were also pooled undiluted in equal volume to normalized samples (*Verkuil et al., 2022*) and submitted for amplicon sequencing. High-throughput amplicon sequencing (HTAS) targeting ~50,000 reads per sample for all samples was conducted on an Illumina® Mi-Seq platform (2 × 250 bp paired end reads) at Azenta Life Sciences (South Plainfield, NJ, USA).

### Sequence read assembly, filtering, and processing

The main goal of our study was to use our hardware system to identify bacterial, plant, and vertebrate taxa are represented in airborne DNA in the atmosphere in our three-dimensional study area. To permit high taxonomic resolution considering low reference database coverage for taxa in the air column, particularly for vertebrates, we used shorter barcode sequences and clustered reads into molecular operational taxonomic unit (OTU) assignments. The bioinformatics pipeline and sequence reference database combinations used were determined by appropriateness for the taxon (and marker) of interest. For example, at present the mBRAVE platform/BOLD database does not support bacterial 16S analyses using the SILVA database and there are very few ITS records on BOLD. Sequencing data for each primer pair were processed separately. FASTQ files for flights and pooled negative controls were processed through the bioinformatic pipelines described below. All raw HTAS datasets were deposited in the NCBI SRA database under BioProject PRJNA906994.

### Bioinformatics pipeline for bacterial *16S* amplicons

Demultiplexed paired end FASTQ files were globally processed using LotuS version 2.21 (*Özkurt et al., 2022*; *Hildebrand et al., 2014*) on the EU Galaxy server at https://usegalaxy.eu/ (*The Galaxy Community, 2022*) with Poisson binomial model based read filtering (*Puente-Sánchez, Aguirre & Parro, 2016*). OTUs were clustered using VSEARCH version
2.21.2 (*Rognes et al., 2016*) with a 2:1 dereplication filter. Taxonomic assignment was performed against the SILVA SSU rDNA reference database release 132 for bacteria (*Quast et al., 2012*) with LCA coverage of 99%, *de novo* and reference database chimera checking and removal, the LULU algorithm to merge OTUs based on their occurrence, and minimap2 version 2.24 (*Li, 2018*) to map back reads to OTUs and exclude off-target alignments. The LotusS2 pipeline offers the advantage of retaining some discarded sequence data when possible with read backmapping and seed extension steps, since using an excessively strict read filter can decrease sensitivity for low-abundance amplicons in the air column by artificially reducing sequencing depth.

### Bioinformatics pipeline for plant *ITS2* amplicons

Demultiplexed paired end FASTQ files were uploaded to the Multiplex Barcode Research And Visualization Environment (mBRAVE) platform (http://www.mbrave.net) for QC trimming, filtering, paired end merging, and OTU bin assignment using the following parameters to maximize information content: trim front: 50 bp, trim end: 50 bp, trim length: 550 bp, min QV: 0, min length: 100 bp, max bases with low QV (<20): 75.0%, max bases with ultra-low QV (<10): 75.0%, ID distance threshold: 10.0%, exclude from OTU threshold: 3.0%, min OTU size: one, OTU threshold: 2.0%, paired end merging with assembler min overlap: 10 bp, and assembler max substitution: 5 bp. Processed reads were screened against the SYS-CRLPLANTITS system reference library of BOLD (Barcode of Life Database) sequences.

As ITS records on BOLD are limited, FASTQ files were also globally processed using LotuS version 2.21 (*Özkurt et al., 2022*; *Hildebrand et al., 2014*) on the EU Galaxy server with Poisson binomial model based read filtering (*Puente-Sánchez, Aguirre & Parro, 2016*). ITS amplicons were clustered into OTUs with CD-HIT version 4.8.1 (*Fu et al., 2012*), global alignment and *de novo* and reference-based chimera detection were implemented with VSEARCH version 2.21.2 (*Rognes et al., 2016*) and UCHIME2 version 11 (*Edgar et al., 2011*) against the UNITE ITS chimera database (*Nilsson et al., 2015*), followed by off-target phiX genome removal (*Bedarf et al., 2021*) and exclusion of off-target alignments using minimap2 version 2.24 (*Li, 2018*). OTU sequences were filtered by default with ITSx (*Bengtsson-Palme et al., 2013*), which identifies likely ITS2 and full-length ITS sequences and discards sequences not within the confidence interval (*Özkurt et al., 2022*). Reads were screened against the comprehensive PLANTITS reference database (*Banchi et al., 2020b*) for plant ITS2 and the UNITE ITS taxonomical reference database (*Kõljalg et al., 2013*) for plant and fungi ITS matches.

### Bioinformatics pipeline for vertebrate *COI* amplicons

Demultiplexed FASTQ files were uploaded to the mBRAVE platform for filtering, dereplication, denoising, identification, and OTU assignment. The following parameters were set to maximize information content: trim front = 38 bp, trim end = 26 bp, trim length = 500 bp, min QV filter = 0, min length = 100 bp, max bases with low (<20) QV = 75%, max bases with ultra-low QV (<10) = 75%, ID threshold = 10%, exclude from OTU at 10%, min OTU size = 1, and OTU threshold = 2%. Paired end reads were

assembled with a minimum overlap of 20 bp and max substitution of 5 bp and then screened against SYS-CRLCHORDATA, SYS-CRLAVES, SYS-CRLINSECTA, and SYS-CRLPROTISTA *COI* system reference libraries.

The demultiplexed FASTQ files were also uploaded to the Galaxy Europe server (https://usegalaxy.eu/) using the DADA2 pipeline to explore *COI* amplicon sequence variants (ASVs) potentially not detected by OTU binning in mBRAVE. Briefly, we used *cutadapt* version 4.0 with Python 3.9.12 and the commands *filterAndTrim()* to remove primers, quality trim and filter reads, *learnErrors()* to generate an error model of our data, *dada()* to dereplicate sequences and infer ASVs on both forward and reverse reads independently, *mergePairs()* to merge forward and reverse reads and further refine ASVs, *makeSequenceTable()* to generate count tables, *removeBimeraDenovo()* to screen for and remove chimeras, and *IdTaxa()* to assign taxonomy. Taxonomic assignment was performed on ASVs using an in-house database of COI-5P reference sequences cut with the AquaF2/Vr1d primer pair in SnapGene® version 6.1 (GSL Biotech, San Diego, CA, USA). DADA2 ASVs were also queried against the NCBI GenBank database of RefSeq mitochondrial sequences using megablast BLASTN 2.10.1+ (*Zhang et al., 2000*). We analyzed Blast output files in MEGAN Community Edition (version 6.21.7) with a weighted LCA with 80% coverage, top percent of 10, and minimum score of 150 (*Huson et al., 2007*). Taxonomic assignment to species was performed if the sequence had 100% identity match to the NCBI reference.

## Global taxonomic profiles and statistical analyses

Filtered taxonomic tables were generated following removal of a subset of OTUs identified from the pooled extraction negative and no-template PCR control. OTU tables for each biological entity (bacteria, plant, and vertebrate) were imported into the Microbiome Explorer Namco version 1.1, an R shiny application (*Dietrich et al., 2022*) following normalization to generate taxonomic profiles and stacked bar plots. We also performed α- and β-diversity statistical analyses and generated ordination plots using the Namco R shiny app.

To estimate alpha (α) diversity for bacteria, plant, and vertebrate OTU assemblages within samples, Shannon and Simpson indices and richness were calculated with their effective indices to account for nonlinearity of the original index. To examine the influence of height (altitude), wind speed, or time as possible explanatory variables, we performed pairwise nonparametric Wilcoxon tests using the SIAMCAT R-package in Namco to compare OTU assemblages between flight samples, as well as Kruskal–Wallis tests (*Kruskal & Wallis, 1952*) for multiple group comparisons with $P$-value significance cutoff of $\leq 0.05$ and Bonferroni multiple testing correction.

To assess beta (β) diversity among samples, scatterplots using Principal Coordinate Analysis (PCoA) classical multidimensional scaling based on Bray-Curtis dissimilarity distance matrices were generated to explore clustering (*Bray & Curtis, 1957*). Flight samples ordinated closer together would represent OTU communities more similar in sequence composition than those further away. To determine the effects of flight, month/season, and altitude on the composition of the airborne OTU communities (read count

data), we performed Permutational Multivariate Analysis of Variance (PERMANOVA) associated with classical multidimensional scaling to test for significant differences between each set of flight samples. We considered $P$-values $\leq 0.05$ indicative that the null hypothesis ($H_0$: samples from all groups are drawn from the same distribution) should be rejected.

## RESULTS

Our airplane-mounted hardware system collected airborne eDNA from bacteria (*16S*), plants (*ITS*), and mammals and birds (*COI*), hundreds and thousands of meters into the atmosphere with levels of diversity representative of the survey area. Overall, HTAS of the six flight samples representing ~10,800 L (10.8 m$^3$) total air volume filtered yielded 2,024,926 sequence reads (1,012 Mb) with a mean quality score of 34.4 and 86.5% of bases with a quality score $\geq 30$ (File S2.Tables 1–3). OTU taxonomic identifications made for all biological entities (bacteria, plants, and vertebrates) are provided in File S2.Tables 4–6. Our hardware consistently captured DNA from prokaryotes (bacteria) and eukaryotes including Pinophyta (conifers), Magnoliophyta (flowering plants), and Chordata (Mammalia and Aves) filtered from the atmosphere (Fig. 5, Data S1.3, File S2).

The bacterial common core detected using our high-integrity capture system was Actinobacteria, Cyanobacteria, Firmicutes, and Proteobacteria (Fig. 5; Data S1.3). Less commonly identified bacterial phyla in the air samples included Bacteroidota, Deinococcota, and Verrucomicrobiota (Fig. 5). Dominant bacterial families in Actinobacteria were Intrasporangiaceae and Kineosporiaceae. Chrooococcidiopsaceae was the most abundant family in Cyanobacteria. Streptococcaceae and Lachnospiraceae were abundant families in Firmicutes. Of the Proteobacteria, Beijerinckiaceae, Burkholderiaceae, Pseudomonadaeceae, Rhizobiaceae, Rhodobacteraceae, and Xanthobacteraceae were the most abundant families detected (Fig. 6). Bacteria enriched at all altitudes (300, 1,200, and 2,500 m) in May included *Cupriavidus*, *Methylobacterium*, and *Pseudomonas*; *Streptococcus*, *Kineosporia*, and *Ligilactobacillus* were also amongst the most abundant genera detected in May (Table 1). Bacteria enriched at all altitudes (300, 1,200, and 2,500 m) in June included *Bradyrhizobium*, *Kineosporia*, *Methylobacterium* and *Streptococcus*. *Cuprividus* and *Phenylobacterium* were amongst the most abundant genera detected in June (Table 1).

Of the plant genera identified, *Pinus* (pine) was most abundant at 300 m in May, while 1,200 and 2,500 m altitudes were dominated by *Quercus* (oak) (Fig. 5; Fig. 7). *Agrostis* (bentgrass) and to a lesser extent *Pinus, Festuca* (fescue) and *Juncus* (rushes) were abundant genera detected this day at 2,500 m. *Pinus* was detected at all altitudes in May (detection at 1,200 m was revealed by Sanger sequencing) (Fig. 5; Data S1.4), which was expected given collections were performed during a seasonal pine pollen event in the springtime that ranged roughly from late March to mid-May. Interestingly, plant taxa detected at two or more altitudes were always detected at 2,500 m. While most of the plants we detected are aeroallergens, we also detected *Lamium* (deadnettle), an invasive weedy species with antimicrobial and anti-inflammatory properties (Fig. 6; File S2).

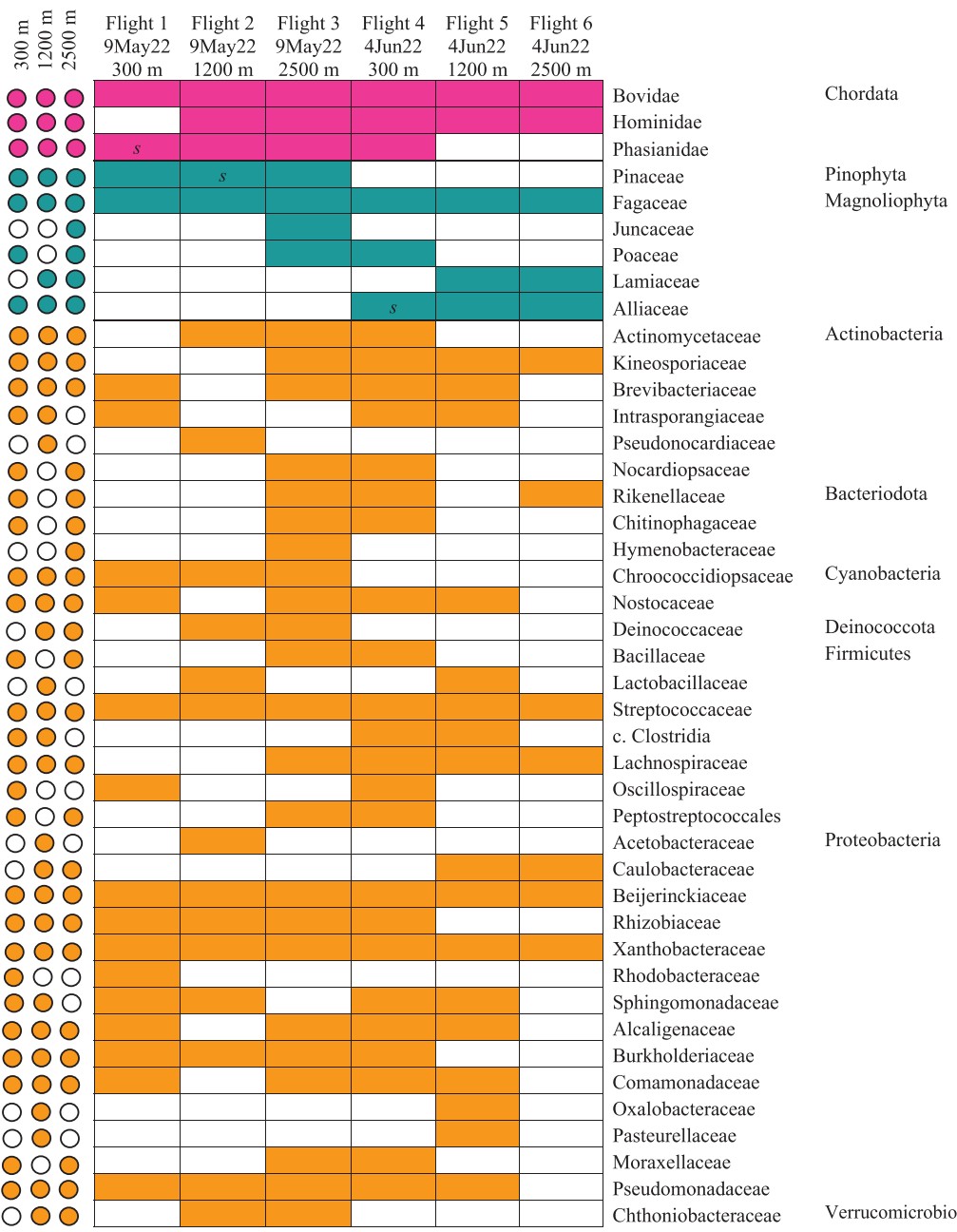

**Figure 5 Heatmap of taxonomic matrices representing presence/absence of each biological group (vertebrates, plants, bacteria) at and above the PBL.** Taxa are represented at the family or closest taxonomic level. Quick-reference coloured circles at left represent detection at that altitude. The three samples marked *s* were inconclusive by metabarcoding but confirmed through Sanger sequencing.

We detected airborne eDNA from mammals (Bovidae and Hominidae) and birds (Phasianidae) in the atmosphere (Figs. 5 and 6). Domesticated chicken (*Gallus gallus*) and cow (*Bos taurus*) as well as human were detected up to high altitude (2,500 m) (Fig. 6). No vertebrate sequence reads were detected in the pooled negative controls.

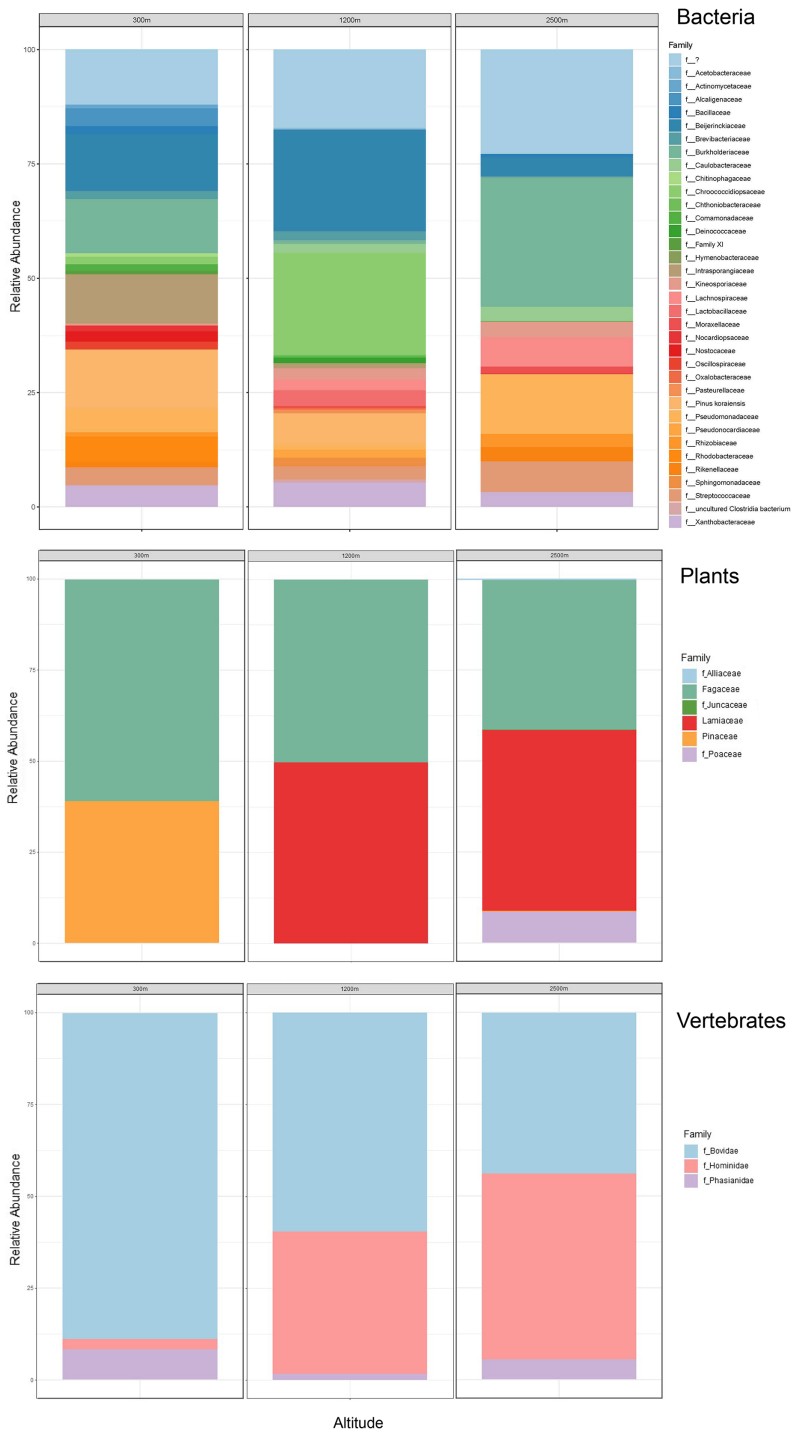

**Figure 6 Taxonomic composition and relative abundance of bacteria, plant, and vertebrate airborne DNA detected by aircraft surveys at 300, 1,200, and 2,500 m.** Taxonomy across biological entities is shown at the family level.

Airborne eDNA at all altitudes (300, 1,200, and 2,500 m) included Vertebrates: Bovidae, Hominidae, and Phasianidae; Plants: Pinaceae, Fagaceae, and Alliaceae; and Bacteria: Actinomycetaceae, Kineosporiaceae, Brevibacteriaceae, Chroococcidiopsaceae,

**Table 1  Top five most abundant genera at each altitude.**

| Taxon | Altitude (m) | 9-May Genus | 9-May Relative abundance (%) | 4-June Genus | 4-June Relative abundance (%) |
|---|---|---|---|---|---|
| Bacteria | 300 | *Cupriavidus* | 22.58 | *Methylobacterium* | 20.03 |
| | 300 | *Pseudomonas* | 9.68 | *Bradyrhizobium* | 4.36 |
| | 300 | *Streptococcus* | 6.45 | *Streptococcus* | 1.37 |
| | 300 | *Methylobacterium* | 4.84 | *Cupriavidus* | 1.17 |
| | 300 | *Chroococidiopsis* | 3.23 | *Kineosporia* | 0.85 |
| | 1,200 | *Chroococidiopsis* | 44.64 | *Methylobacterium* | 20.02 |
| | 1,200 | *Methylobacterium* | 24.36 | *Bradyrhizobium* | 10.48 |
| | 1,200 | *Ligilactobacillus* | 1.97 | *Streptococcus* | 5.65 |
| | 1,200 | *Cupriavidus* | 1.63 | *Kineosporia* | 4.95 |
| | 1,200 | *Pseudomonas* | 0.47 | *Phenylobacterium* | 3.89 |
| | 2,500 | *Cupriavidus* | 56.57 | *Streptococcus* | 12.50 |
| | 2,500 | *Pseudomonas* | 26.06 | *Bradyrhizobium* | 6.25 |
| | 2,500 | *Methylobacterium* | 2.07 | *Kineosporia* | 6.25 |
| | 2,500 | *Streptococcus* | 0.94 | *Methylobacterium* | 6.25 |
| | 2,500 | *Kineosporia* | 0.43 | *Phenylobacterium* | 6.25 |
| Plant | 300 | *Pinus* | 77.78 | *Quercus* | 99.81 |
| | 300 | *Quercus* | 22.22 | *Agrostis* | 0.19 |
| | 1,200 | *Quercus* | 100.00 | *Lamium* | 99.30 |
| | 1,200 | | | *Quercus* | 0.46 |
| | 1,200 | | | *Allium* | 0.23 |
| | 2,500 | *Quercus* | 82.25 | *Lamium* | 99.46 |
| | 2,500 | *Agrostis* | 17.03 | *Allium* | 0.52 |
| | 2,500 | *Pinus* | 0.38 | | |
| | 2,500 | *Festuca* | 0.21 | | |
| | 2,500 | *Juncus* | 0.10 | | |
| Vertebrate | 300 | *Bos* | 90.74 | *Bos* | 75.00 |
| | 300 | *Bos/Bubalis* | 9.25 | *Gallus* | 16.67 |
| | 300 | | | *Homo* | 5.56 |
| | 300 | | | *Bos/Bubalis* | 2.78 |
| | 1,200 | *Bos* | 89.83 | *Homo* | 74.00 |
| | 1,200 | *Bos/Bubalis* | 3.39 | *Bos* | 26.00 |
| | 1,200 | *Gallus* | 3.39 | | |
| | 1,200 | *Homo* | 3.39 | | |
| | 2,500 | *Bos* | 65.93 | *Homo* | 78.19 |
| | 2,500 | *Homo* | 23.08 | *Bos* | 21.65 |
| | 2,500 | *Gallus* | 10.99 | | |

**Note:**
Numbers are relative abundances (%) from high-throughput amplicon sequencing.

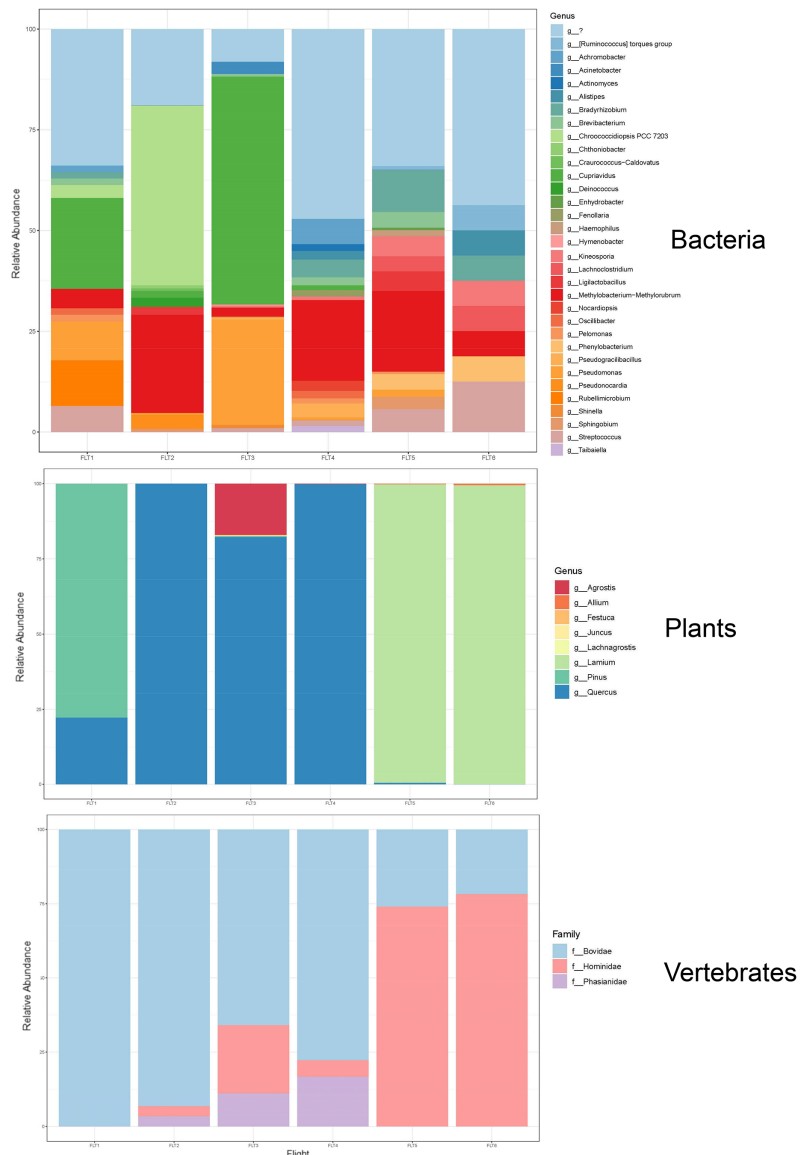

**Figure 7** **Taxonomic composition and relative abundance of bacteria, plant, and vertebrate airborne DNA on six research survey flights.** Taxonomy shown at the level of genus except for vertebrates (family).

Nostocaceae, Streptococcaceae, Lachnospiraceae, Beijerinckiaceae, Rhizobiaceae, Xanthobacteraceae, Alcaligenaceae, Burkholderiaceae, Comamonadaceae, and Pseudomonadaceae (Fig. 5; Fig. 6). Of these, Bovidae (*Bos taurus*), Hominidae (*Homo sapiens*), Fagaceae (*Quercus*), Streptococcaceae (*Streptococcus*), Beijerinckiaceae (*Methylobacterium-Methylorubrum*), and Xanthobacteraceae (*Bradyrhizobium*) were detected on all six sampling flights in May and June (Fig. 5). The flight with the most vertebrate, plant, and bacterial detections was at 2,500 m in May on Flight #3 which was also our lowest volume of air sampled. Environmental DNA from three vertebrate taxa, four plant taxa, and 23 bacterial taxa were co-captured on this flight (a similar number of

bacterial OTUs were detected at 300 m on Flight #4 in June). We found no significant difference in mean detection across days at the same altitude for vertebrates and plants, although our results suggest that higher altitudes may increase the probability of detecting rare species.

To understand the influence of flight altitude on the diversity of bacterial, plant, and vertebrate DNA assemblages captured with our sampler, we calculated the Shannon and Simpson indices of α-diversity. For all altitudes, bacteria assemblages, followed by plants, had the highest Shannon indices for α-diversity and greatest number of OTUs, highlighting more diverse community structure, although Wilcoxon tests were not significant (Data S1.3 Table 1; Data S2). OTU (species) richness was consistently highest at 300 m for both bacteria and plants. OTU richness was found to decrease with increasing altitude for bacteria and for plants, except for plant DNA sampled at 1,200 m (in line with rarefaction curves for Flight #5) (Data S1.3 Table 1). Overall, fewer OTUs dominated the vertebrate community assemblages with less diversity and richness. The α-diversity of detected mammalian and avian OTUs increased with altitude up to 2,500 m (Data S1.3 Fig. 3).

We used Bray-Curtis PCoA scatterplots to explore β-diversity of the OTU assemblages. For bacterial OTUs, communities were most similar in sequence composition on Flights #2, #4, and #5 (spanning across flights in May and June), and the bacterial assemblage from Flight #3 (2,500 m in May) was the most dissimilar (Data S1.3 Fig. 4). Plant OTU communities from samples on Flights #1 and #2 (May) were similar in sequence composition, as were samples on Flights #5 and #6 (June). Flights #3 and #4 (spanning May and June flights) captured similar OTU communities for plants. Regarding vertebrate OTU assemblages, flight samples from May resemble each other in terms of sequence composition, as do those from June flights. No significant differences between groups were revealed by PERMANOVA (Data S1.3).

Atmospheric conditions for survey flights in May were relatively calm with no gusts (mean wind velocity 4.6 kts) and visible aerosols as smoke plumes (Data S1.2). However, the weather during all June flights was characterized by wind vector shifts, gusty conditions (19+ kts), convection, updrafts, and turbulence (Data S1.2). Visible moisture as clouds was observed on all flights in June. Our flights were conducted during published observations of recent high surface pressure systems and more stable, settled weather (Data S1.2).

HYSPLIT 48-h back trajectories for the modelled heights were also not congruent between May flights and June flights, nor between air masses at different heights sampled on the same day (Fig. 8). In May, the air mass that circulated at the highest altitude (2,500 m) originated from the north in Canada near the Arctic circle moving south into the US; the air mass that circulated at 300 m originated from the Gulf and moved along the northeast coast of the US; and the air mass circulating at 1,200 m originated from the northeast US. Notably, the greatest diversity of plant species was detected at 2,500 m in May on Flight #3 through the air mass transported from the North Arctic, including Junacaceae (Magnoliophyta) and *Agrostis* (Poaceae). Bacterial family Hymenobacteraceae was detected exclusively from this flight, as was an abundance of *Cupravidus* (Fig. 7). In June, the air mass that circulated at 2,500 m originated from the Gulf and travelled

northeast, and the air masses comprising the sampled altitudes of 300 and 1,200 m travelled to the study area from the Great Lakes and midwestern US, respectively. Several bacterial families detected on flights at 300 and 1,200 m this day were not captured flying through air mass at 2,500 m, suggesting the possibility that atmospheric conditions such as air parcel transport may contribute to composition of DNA assemblages.

## Methodological considerations

This study used active countermeasures to eliminate foreign DNA contaminants, supplemented with comprehensive negative controls. The species with highest abundance (most reads) in the negative controls, *Arabidopsis thaliana* and *Escherichia coli*, are organisms we commonly use in the adjacent teaching lab. Species identified in the negative control sequencing pool (File S2.Tables 3–6) were discarded from samples as probable false positive detections. This conservative approach to manage the risk of reporting false-positives means that we likely underestimated presence/diversity of some genera, particularly *Brassica* and *Escherichia*, which have been reported in atmospheric DNA samples (*Sánchez-Parra et al., 2021*; *Jaing et al., 2020*).

While vertebrate DNA can be a common contaminant in labs (*Leonard et al., 2007*), it was absent from our pooled negative control. Here, the comprehensive controls and context of our study are key. All negative controls were free of all vertebrate/mammal reads, including human, indicating the robustness of our approach to guard against lab contaminants. Bovine serum albumin (BSA) was not an additive in our PCR. All vertebrate species we detected are plentiful in our study region, and facilities housing them (as well as the animals themselves) are visible from all altitudes flown. The most visible ground features whilst airborne are livestock and poultry production facilities and agricultural land (Fig. 3; File S1). Industrial buildings such as wastewater treatment plants and construction activity (grating for housing and other urban development) as well as human dwellings are also visible from the aircraft in the study area. Air detections at ground level of cow and chicken DNA in captive environments have been reported (*Lynggaard et al., 2022*; *Clare et al., 2022*), and along with horse and pig, represented up to one third of sequences identified (*Clare et al., 2022*). Likewise, one of the pioneering studies to establish the field of eDNA in aquatic systems was predicated on detections of human, cow, pig, and sheep from feces in groundwater using mitochondrial markers (*Martellini, Payment & Villemur, 2005*).

We took several steps to optimize collection efficiency and yield, even though our goal was not to exhaustively quantify biodiversity in flight samples using DNA copy number or cell counts or additional markers for taxa such as fungi or protists. All air sampled by our probe was filtered to capture nucleic acids and could not bypass the collection media. Our sampler does not dilute samples in a collection liquid which could allow loss through evaporation, bubbling and reaerosolization, or sticking to the walls of the sampler. Further, we engineered the isolation chamber with check valves to minimize sample loss until laboratory processing. Overall, elution efficiencies after processing using the methodology reported herein were typically 5 to 100 ng/$\mu$L of DNA per filter, which provided

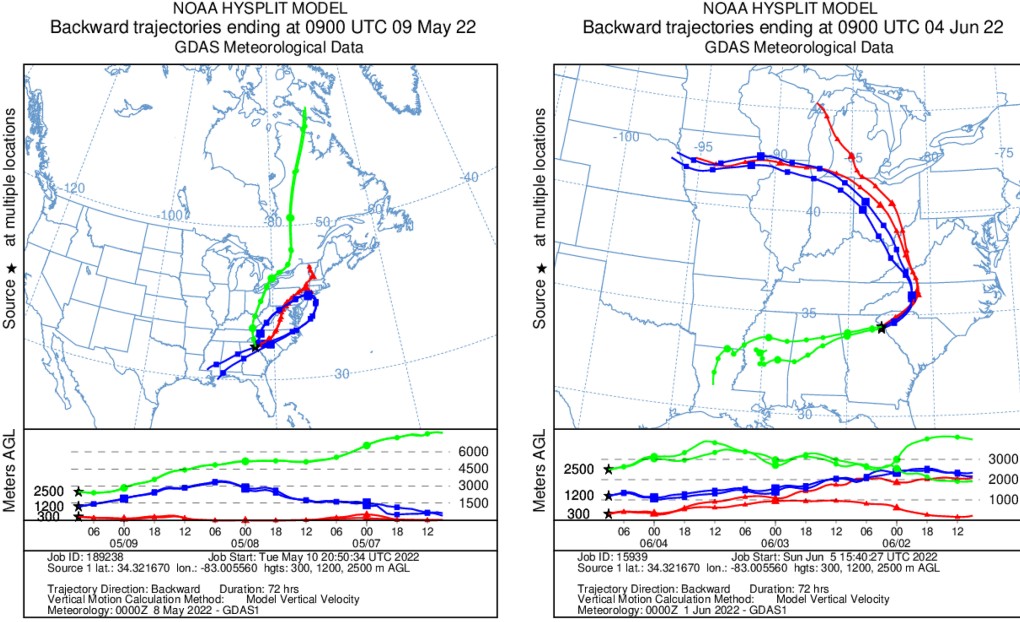

**Figure 8 Kinematic back trajectories modelling air mass transport histories for flights in May and June.** Representation of backward 72 h forecast trajectories from the survey area grid location of 34.32 N, −83.01 W at 300, 1,200, and 2,500 m above ground level using HYSPLIT and GDAS (Global Data Assimilation System) meteorological data. The representation is superimposed on a physical map and displayed as a schematic for three flights per day at different altitudes. Map credit: © NOAA.

satisfactory material for sequencing and coverage of microbial diversity including OTUs with low relative abundances.

We met several constraints due to technical restrictions of our current research platform, a light aircraft, including max gross weight and center of gravity limitations plus a scientific payload of 15.8 kg for this airplane. In this aircraft we could only install up to two sampling probes. However, our hardware system is designed to collect as many samples per flight as the payload of the aircraft or platform can support. We designed modular probes to be installable in series across larger research platforms. We were also limited by the aircraft's power capacity to supply the vacuum pumps. We have access to 12 V power and 20 A but ideally would need 110 V power or more to drive larger vacuum pumps which can pull a greater volume of air. Further, we are restricted by federal regulations that the aircraft total electrical load must be inferior to 80% of its charging system capacity. Here, we allowed maximum flow possible to ensure optimal capture and retention of nucleic acids from bioaerosols of multiple types. In future, we can increase the rate of air flow by approximately 30% with the addition of a second vacuum pump.

Despite these limitations, using a light aircraft as a genetic research platform also presents advantages. Fuel burn is less than 19 L·h$^{-1}$ (5 gph) at economy cruise for the aircraft used in our study. The operating costs are US$250 per flight hour (including crew of two and fuel), whereas the operating costs of the DC-8 and Gulfstream C-20A (GIII) are currently $6,500 and $3,000 per flight hour, respectively (NASA ASP Flight Request

Planning Tool, https://airbornescience.nasa.gov/content/Costing_Support, accessed 22 November 2022). Our pump-driven system does not rely on ram air, so it is efficient at slow airspeeds or stationary ground or flight regimes, such as a hovering helicopter or UAV. The aircraft cabin is not pressurized (and our probe is designed to remain outside the aircraft), so we do not have to contend with a pressure differential from inside the aircraft cabin and external ambient air pressure. Samplers which will meet the challenges of bioaerosol sampling in extreme environments should be lightweight (*Mainelis, 2020*), and our system fills that niche. While light aircraft are suboptimal for sampling altitudes beyond about 20,000 feet, they can extend beyond a fixed-point observation or stationary collection and probe the entire depth of the planetary boundary layer (*Lenschow, 1986*). Our study highlights that light aircraft can serve as a platform to conduct airborne genetic research with spatio-temporal resolution at a scalable level in open spaces. Taken together, our airborne campaign to detect genetic material in the sky is scalable and cross-platform integration is possible, which would be advantageous for mission flexibility and diverse applications.

## DISCUSSION

Aerosolized DNA from bacteria, plants, and vertebrates is present at large scale, thousands of meters vertically into the troposphere. Using novel aerial genetic surveys, we detected airborne DNA at each height sampled up to high altitude (2,500 m above ground level). The air mass back trajectories indicate our flights passed through air parcels (and presumably particles) with different transport histories, suggesting that the large-scale genetic presence of eDNA in bioaerosols throughout the PBL remains detectable despite flying through different air sources. This is supported by other researchers who reported that metazoan and plant taxa are airborne independent of meteorological conditions, and few bacteria display weather-related modulation of the air microbiome (*de Groot et al., 2021*).

Our work supports and extends previous studies of airborne DNA (*e.g.*, in bacteria: *Sánchez-Parra et al., 2021*; *Jaing et al., 2020*), in plants: *Craine et al., 2017*; *Damialis et al., 2017*; *Sánchez-Parra et al., 2021*; *Folloni et al., 2012*) by revealing widespread presence of DNA in the atmosphere. We identified mammal and avian species from airborne DNA like other groups (*Clare et al., 2021*; *Clare et al., 2022*; *Lynggaard et al., 2022*; *Serrao et al., 2021*; *Aalismail et al., 2021*), but at a much greater distance and larger scale. Instead of using a passive sampler or capturing DNA from dust, we actively filtered a measurable volume of air at altitude for genetic material. It is remarkable how large-scale the dispersal is (at least vertically) considering strong evidence from aquatic studies that environmental DNA is localized in water systems (*Littlefair et al., 2021*). We found no indication that terrestrial DNA is less abundant or readily detectable at high altitude than near the surface. However, our evidence does indicate that the ground-level major sources of bioaerosols, coupled with atmospheric conditions, play an important role in the composition of multi-taxa air eDNA assemblages. Markedly rural-agricultural land peppered with mechanically ventilated poultry houses and cattle pastures dominated the area we overflew on these research flights, yet even with interspersed forests below, no wildlife species were detected

in the animal DNA captured aloft. Indeed, the atmosphere culls and everything is not everywhere (*Griffin et al., 2011*). This suggests an important role for aerosolization efficacy of emissions sources in modulating air eDNA assemblages at higher altitudes in the troposphere.

## The bacterial aerobiome

Several bacterial orders detected in our study have been reported in rural, suburban, and mountain air environments previously (*Ruiz-Gil et al., 2020*), including Pseudomonadales (*Pseudomonas* and *Acinetobacter*), Rhizobiales (*Bradyrhizobium* and *Methylobacterium*), and Cytophagales (*Hymenobacter*). Staphylococcaceae, Oxalobacteraceae, Lachnospiraceae, Burkholderiaceae, and Moraxellaceae were also detected by *Smith et al. (2018)* in airborne surveys of the western US. *Phenylobacteria* have been isolated from the soil and rhizosphere, beach sand, pesticide manufacturing sludge, and water purifier reservoirs (*Chu et al., 2015*; *Farh et al., 2016*; *Jo et al., 2016*). One of the most abundant genera detected in this study, Cyanobacteria genus *Chroococcidiopsis*, is an extremophile able to thrive in a wide variety of harsh environments including space and Martian conditions (*Billi, 2019*; *Napoli et al., 2022*). Interestingly, despite being a ubiquitous group of soil bacteria, many reports of airborne Cyanobacteria come from coastal, not inland, environments (*Graham et al., 2018*).

Many bacteria we identified are human fecal microbiota (*Martínez, Muller & Walter, 2013*). Dominance of Actinobacteria, Clostridiales, and Bacteriodes are associated with livestock feces (*Wei et al., 2019*). We detected many anaerobic bacteria (*e.g.*, Bacteriodales, *Clostridium, Actinomyces, Lactobacillus*) in our study, suggesting a non-air source and the possibility that anaerobic digestion of sewage and sludge from wastewater treatment plants contributes to the studied microbiome (*Cyprowski et al., 2018*). Wastewater treatment aeration and agitation systems as well as livestock and poultry facility aeration systems are known to have high bacterial concentrations (*Yang et al., 2019*). We identified spore-forming bacteria such as Clostridiales and Bacillales, but other microbes we detected do not form spores (*e.g.*, Burkholderiales: *Pelomonas*). We detected numerous bacteria that thrive in clouds during our aerial surveys, including Burkholderiales, *Methylobacterium*, and *Sphingomonas*, which have been reported to use atmospheric components as nutrient sources (*Amato et al., 2017*). Size dependent dispersion could contribute to higher α-diversity with altitude for bacterial assemblages, and perhaps also for smaller plant pollen which may reside in clouds as condensation nuclei (*Amato et al., 2017*). In addition, we captured several bacteria potentially useful for bioprospecting, including *Methylobacterium*, which can reduce environmental contamination, degrade toxic compounds, and tolerance to heavy metals in plants (*Dourado et al., 2015*) and *Pseudomonas, Bradyrhizobium*, and *Hymenobacter*, all known sources of natural pigments (*Narsing Rao, Xiao & Li, 2017*).

The Actinobacteria, Bacteroides, and Cyanobacteria classes we detected are soil-associated bacteria, with a few exceptions. Since we used a soil extraction protocol, it is not surprising that this a common source medium; our work extends that of *Hermans, Buckley & Lear (2018)* who found that soil extraction protocols broadly isolate nucleic

acids from multiple taxa collected from various environmental sources but did not examine air. *Brevibacterium oceani* is associated with deep sea ocean sediment which has not been reported previously in the air, although genus *Brevibacterium* is dominant in chicken farm excreta (*Bindari et al., 2021*). The Bacteroides genus *Alstipes* is isolated from clinical samples and human feces, suggesting a possible hospital or landfill source. Bacilli and Clostridia detected are almost exclusively associated with poultry, bovine, human, porcine gastrointestinal microbiota derived from animal manure or fertilizer. We isolated several pathogens of class Gammaproteobacteria (*e.g.*, *Haemophilus* and *Acinetobacter baumannii*) of noteworthy clinical relevance that are known hospital-associated bacteria. *Haemophilus*, part of the human salivary microbiome, is associated with a variety of airborne infections. Environmental strains of *A. baumannii* have also been reported in ancient paleosol (*Hrenovic et al., 2014*). Other Gammaproteobacteria we detected are environmental bacteria common in soil and wastewater.

Our sampler also detected less commonly reported airborne species such as Verrucomicrobiota (*Mbareche et al., 2018*). Verrucomicrobiota is a widespread phylum of free-living bacteria that contains few described species, typically isolated from groundwater, soil, hot springs, leaf matter, and human feces (*Hou et al., 2008*), specialist consumers of complex polysaccharides during algae blooms (*Orellana et al., 2022*). We only found one previous report of Verrucomicrobiota family Chthoniobacteraceae isolated from air up to 130 m in Spain (*Sánchez-Parra et al., 2021*). We detected microbes not formally identified in the air previously, such as *Brevibacterium oceani*. We also detected *Taibaiella*, a Gram negative, aerobic bacterium commonly isolated from soil, weeds, and grasses but rarely reported in air.

## The plant aerobiome

Plant DNA was detected up to 2,500 m from various deciduous taxa including Fagaceae, (*Quercus*), Pinaceae *(Picea/Pinus)*, and *Allium sativum*. The most common plant families across flights were Fagaceae, Lamiaceae (*Lamium* or deadnettle is an agricultural weed), and Pinaceae. Pinaceae, Fagaceae, and Poaceae are common detections between this study and Europe (*Sánchez-Parra et al., 2021*), although differences in the communities detected are expected given the habitat differences. Here, we provide the first report of the herb *A. sativum* L. (garlic) detected hundreds and thousands of meters into the atmosphere (at all altitudes on June flights). While previous reports of sequenced garlic DNA captured at altitude do not exist, others have identified *Quercus* and Pinaceae sequences in dust samples (*Craine et al., 2017*; *Damialis et al., 2017*) and Pinaceae in air samples (*Sánchez-Parra et al., 2021*) including off-target detection of *Pinus* at 10,000 ft MSL over a forested area of the Sierra Nevada mountains (*Jaing et al., 2020*). Transport modelling of deciduous hardwood pollen in air (*Skjøth et al., 2007*) suggests the possibility of large-scale aeroallergen exposure (*Longhi et al., 2009*). In addition to aerosolized plant matter from urban and suburban residences, agroecosystems, and fertilizer application, it is also possible that some of the plant detections such as *Festuca* (fescue) are from aerosolized manure containing digested plant matter, underscoring the connectivity of air DNA assemblages captured from the atmosphere.

We selected ITS primers to target plants specifically (not fungi). However, *(Neo) Ascomycota* fungal DNA amplified by our plant specific ITS primers was present in flight samples at 300, 1,200, and 2,500 m (as well as the negative control) (File S2). *Ascomycota* are plant pathogens involved in nutrient recycling in the ecosystem (*Golzar et al., 2019*). Our detection is supported by *Sánchez-Parra et al. (2021)* who captured *Ascochyta* at 1.5, 130, and 500 m (but not in their 1,000 m sampling) in Spain. Spores, often reported to resist environmental stress and survive atmospheric transport, may have facilitated detection of this fungus in the air column (*Griffin & Kellogg, 2004*).

### High-altitude aerosolization of terrestrial-associated air eDNA

Our airborne genetic surveys reveal that vertebrate eDNA from agricultural sources (cow and chicken) is present at high altitudes in the PBL and above, providing evidence supporting the cow fecal microbiome in the near-surface atmosphere (*Bowers et al., 2013*) that aerodynamic nucleic acid-based bioaerosols from feces, dander, feathers, or bodily fluids can spread throughout the atmosphere. The vertebrate and enteric bacterial DNA we detected likely originates from animal production facilities, fertilized plots, landfills, or wastewater treatment plants found throughout our study area, and perhaps from hospitals and other facilities that aerate biological waste (File S1). Our flights were positioned over hypothesized DNA aerosolization sources (*e.g.*, poultry houses, which constitute a primary driver of the local economy), where cooling fans and mechanically ventilated exhaust systems are used to increase ventilation and remove excess heat, presumably leading to higher DNA emissions (*Guo et al., 2020*; *Zhang et al., 2022*). Our results point to the importance of monitoring agricultural bioaerosols, as poultry manure, litter, feather fragments, skin, microbiota, and related bioaerosols are associated with adverse environmental and health impacts, including acute and chronic respiratory disorders (*Oppliger et al., 2008*; *Viegas et al., 2013*).

The air eDNA we detected shares associations with ground-level emissions sources below or near our study area (*e.g.*, waste treatment activities or agricultural activities such as farming, fertilizer application), which could have important and serious implications for industrial practices (*Jones & Harrison, 2004*). These findings suggest that natural ecosystem processes (*e.g.*, mechanical suspension from lifting mechanisms or wind, pollination of forests) as well as anthropogenic activities (*e.g.*, poultry production, farming, controlled burns, wastewater treatment, hospital waste decontamination) could be important sources of bioaerosols in the atmosphere (*Alsved et al., 2020*).

### Atmospheric impacts on the fate and detection of air eDNA

Our aerial surveys limited the effect of wind vector direction and time of day on airborne eDNA capture. However, wind can aerosolize land-applied biosolids when soils are sandy and dry (*Fécan, Marticorena & Bergametti, 1999*) or wind velocities are greater than 5 m/s (~9.7 kts) (*Baertsch et al., 2007*), reflected in local conditions during Flights #4 and 5 in this study (Data S1.2). On May 9, early morning winds at the surface were from the north over the Appalachian Mountains, which often brings turbulent conditions (authors' observations) to aerosolize DNA. On June 4, winds were again generally out of the north

and characterized by strong gusts and greater velocity than May. Across all research flights, winds were strongest (11 kts at the surface gusting to 19 kts, Data S1.2) during survey Flight #4 in June at 300 m, with a narrow temperature-dewpoint spread and visible moisture as well as observed thermals and light turbulence, which is when we detected *A. sativum* (garlic). In addition, there were two bacterial family detections unique to this flight: *Oxalobacteraceae* and *Pasturellaceae* (*Haemophilus*) (Fig. 5). On Flight #4 we also detected cow, chicken, and human DNA (Fig. 5). These findings suggest that lifting action and atmospheric mixing can influence the movement and detection of airborne eDNA.

Our results reveal widespread presence and connectivity of terrestrial-associated bacterial, plant, and vertebrate DNA up to high altitude for all flights. It is possible that air parcels with different transport history and particles could introduce unique detections. Mechanisms of lift that cause air to rise and fall—such as convergence, convective lift, frontal lift, and orographic lift—could also cause vertical movement and mixing of eDNA in the air. Convergence of air toward a low pressure center generates lift and leads to cloud formation and precipitation. Solar radiation and convection warm air at the surface, which would rise and carry bioaerosols. In a cold front, denser, cold air will undercut and push warm air vertically, which leads to thunderstorms. In a warm front, less dense warm air will rise over cooler air at the surface, which would provide lift for eDNA. In our study area, orographic lift mainly occurs along the Appalachian Mountain range. When wind moves across the Blue Ridge valley (from the south or from the north), it will force air masses at the surface up the side of mountain ridges (orographic lifting from advection), rising until it begins to cool and condense. Inversion layers with eDNA might rise vertically and then sink back to toward the ground and spread horizontally. Any strong force that lifts air or water up into the atmosphere (*e.g.*, a thunderstorm, or updrafts and downdrafts) is likely a transporter for DNA. Thus, we recommend assessment of convective parameters and indices considering lifting action, atmospheric instability, and potential for convection (*i.e.*, the Lifted Index, K index, or Convective Available Potential Energy, CAPE) to predict DNA levels in the air prior to sampling (reviewed in *Kunz (2007)*). Lifting mechanisms may be more dominant forces for mixing eDNA in the atmosphere during the day, since inversion layers typically result at night as warmer air overlies cooler air closer to the surface, but since we minimized the effects of time of day this possibility remains unexplored here.

## Challenges and prospects

At present, collective progress in this emergent area of research is impeded by a severe lack of coverage in global genetic reference databases for species occurring in the air column. Integrative, multi-taxa metabarcoding can provide a fast and less expensive way to explore all Earth's biodiversity in the atmosphere, but marker choices and reference databases for biological entities in the air column will need to be broadened, as they have been for aquatic systems. Reference databases are currently more complete for airborne bacterial sequences compared to animal *COI* and plant *ITS* isolates from air. In some cases, we could

not achieve a species level taxonomic assignment for eukaryotes and there were many OTUs left unassigned. Further, interspecific mitochondrial introgression or hybridization (*e.g.*, hybridization of *Festuca* with other grasses we detected), a single marker approach may not provide sufficient taxonomic resolution particularly when differentiating species complexes. In our study, the degeneracy of the *COI* primers also resulted in amplification of other taxonomic groups in addition to the vertebrates targeted in the atmospheric DNA mixtures (*Deagle et al., 2014*). The BOLD systems database binned singleton and doubleton *COI* sequence reads into OTUs matching flying beetles and dragonflies as well as Cnidaria, and although the former is plausible (*e.g.*, flying insect species are expected in the air) they were discarded as they represented off-target detections (File S2.Table 3). *COI* reads similar to Cnidaria may derive from aerosolized organic fertilizers containing these organisms (*Emadodin et al., 2020*). Together, this highlights that continued genetic marker and reference database development will be necessary to improve identification of biodiversity from the sky.

In this study, we detected DNA at altitude from bacterial and vertebrate families associated with enteric, fecal-oral or airborne transmission routes, and others have captured animal parasites (*e.g.*, fungi and protists) using volumetric impaction sampling 12 m above ground level (*de Groot et al., 2021*). Our airborne detections of pathogenic or clinically relevant microbes common in hospital settings (*e.g.*, *Haemophilus* and *Acinetobacter baumannii*), and plants whose pollen is severely allergenic in the local area (*e.g.*, *Quercus, Agrostis, Festuca, Pinus*), suggest relevance to disease ecology surveillance. Bioaerosols discharged from ground emissions can cause acid rain, radiation, and reduced atmospheric visibility and air quality in addition to harming human, animal, and ecosystem health (*Schmale & Ross, 2015*; *Van Leuken et al., 2016*). There are likely atmospheric conduits for disease (*Prospero, 1999*; *Smith et al., 2018*; *Zhao et al., 2014*), and biomonitoring using DNA at scale could overcome constraints of spatially discrete surveillance to facilitate early disease detection and understand dispersal (*Mahaffee & Stoll, 2016*). Airborne organic matter and bacteria—including genera identified in this study—can become ice or cloud condensation nuclei, influence precipitation and exacerbate climate change, deposit phytopathogens, and in the case of pollen, drive plant adaptation or other population genetic processes (*Schmier & Ebi, 2009*; *Ariya et al., 2009*; *Creamean et al., 2013*; *Pratt et al., 2009*; *Ellstrand, 1992*).

Logical next steps for future studies on the persistence of atmospheric DNA extending our work might ask (1) whether nucleic acid concentrations relate to organismal biomass; (2) what particulate matter captured at altitude provides the genetic material (*e.g.*, feces, skin or hair, whole cells or biofilms, pollen grains, leaves, or soil granules); or (3) how eDNA and eRNA levels compare in the atmosphere. To explore the fate of aerosolized genetic material further, future studies could diffuse "tagged" bioaerosols from a ground source for airborne plume tracking coupled with metagenomics.

Our hardware system successfully captured airborne bacteria, plant, and vertebrate DNA assemblages through air masses sampled at stratified altitudes, but we are unlikely to have achieved the fine-scale resolution that daily or weekly sampling over an extended period or seasons could provide, which motivated us from the outset to design our

technology and method to be cost-effective for longer-term and longer-range continuous monitoring efforts. Future investigations will benefit from being frequent and sustained, as changes in the aerobiome can be rapid and substantial (*Macher, 1999*). Construction of airborne DNA profiles using our hardware to map the wider atmospheric biological community would improve air quality models and forecasts. Such efforts could inform industrial air handling best practices for ventilation systems, or pharmaceutical uses of genetic aerial mapping for targeting areas of need or bioprospecting opportunities.

Overall, our work is a demonstration that environmental DNA from terrestrial-associated bacteria, plants, and vertebrates is detectable in the air up to high altitude using our reusable high integrity capture system and highlights the usefulness of light aircraft in monitoring campaigns. We show the operability of our airborne system to collect DNA from diverse taxa in the same air sample using one workflow noninvasively. With this contribution, we hope to inspire cross-disciplinary collaboration surrounding scalable monitoring of global diversity from the air.

## CONCLUSIONS

Environmental DNA is present at large scale in air thousands of meters above the Earth's surface, and it is now possible to identify multiple taxa from the same sample captured in large airspaces. The airborne metagenomic data acquired with our high-integrity hardware system are relevant to the public health and agriculture sectors and contribute to a broader understanding of atmospheric processes and global ecological impact. However, our results also underscore the need for improved marker choices and reference databases across all domains of life created from the air column. Broadly, this study and the increasing body of aerobiology research highlights that air quality, pollution, pest and infectious agent assessments from various industries (*e.g.*, livestock or infrastructure) would benefit from including airborne metabarcoding surveys in addition to tracking nonbiological chemical pollutants and particulate matter. We aim to establish the foundation for airborne genetic surveillance campaigns from light aircraft and inspire further cross-disciplinary approaches, opening the door for the scientific community to access transformative opportunities surrounding precision biomonitoring in the air at scale.

## ACKNOWLEDGEMENTS

The Galaxy server that was used for some calculations is in part funded by Collaborative Research Centre 992 Medical Epigenetics (DFG grant SFB 992/1 2012) and German Federal Ministry of Education and Research (BMBF grants 031 A538A/A538C RBC, 031L0101B/031L0101C de.NBI-epi, 031L0106 de.STAIR (de.NBI).

### Funding

There was no external funding received for this study.

## Competing Interests

Kimberly L Métris and Jérémy Métris are co-founders of Airborne Science, the company that designed and built the high integrity probe and supporting system used in this work. A patent application has been filed for the high-integrity capture system. The authors and company did not receive financial support or compensation for this study.

## Author Contributions

- Kimberly L. Métris led the study, conceived and designed the experiments, co-invented the sampling device, performed the experiments, collected and analyzed the data, prepared figures and tables, authored and reviewed drafts of the manuscript, and approved the final draft.
- Jérémy Métris designed the experiments, co-invented the sampling device, collected the data, prepared figures, authored and reviewed drafts of the manuscript, and approved the final draft.

## Patent Disclosures

The following patent dependencies were disclosed by the authors:

A patent application has been filed for the high-integrity capture system. Engineering specifics regarding the probe, manifold, and vacuum pump are available from the corresponding author.

## Data Availability

High-throughput amplicon sequencing raw reads are available at the NCBI Sequence Read Archive (SRA) database: PRJNA906994.

## Supplemental Information

Supplemental information for this article can be found online at http://dx.doi.org/10.7717/peerj.15171#supplemental-information.

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
