# Peer review of "Aircraft surveys for air eDNA: probing biodiversity in the sky"

_PeerJ, doi:10.7717/peerj.15171_

## Round 0.1 · original submission · Minor Revisions

This paper is significantly improved from the previous submission. Both reviewers have included a number of suggestions to help improve the paper. In particular, please pay attention to Reviewer 1’s comments about organization, and supplementary material.

Reviewer 1 ·

Basic reporting

I suggest the authors to include the two following references:
1) work done by Aalismail et al 2021 (Diversity and Sources of Airborne Eukaryotic Communities (AEC) in the Global Dust Belt over the Red Sea, Earth Systems and Environment). You could compare your results to the ones from the suggested manuscript.
2)In addition, your findings of detecting pathogenic and non-pathogenic bacteria, plants and vertebrate DNA within the same filter aligns well with what has been suggested previously as a potential use with airborne eDNA (reference Bohmann & Lynggaard 2023, Transforming terrestrial biodiversity surveys using airborne eDNA, Trends in Ecology & Evolution).

Experimental design

Line174- I am aware that you want to patent the device, but it would be helpful for the reader to know what type of filter was used. The pore size, or the class of the filter will determine what particle size is retained and could explain why almost no vertebrates were detected (ie. vertebrate DNA could be free floating or degraded easier than pollen and bacteria, and therefore a filter that targets larger particles would maybe not be good for retaining small particles).
Line 355- I agree that it is important to have this negative control. Please refer to other studies that have done something similar.
Line 444- Not clear why you mention that the amplicons were not fragmented.
Not clear what was the reason for using different bioinformatics approaches with the different taxonomic groups.

Validity of the findings

Dear authors,
In your manuscript “Aircraft surveys for air eDNA: probing biodiversity in the sky” you showcase the use of a new device for sampling DNA in air at different heights in the atmosphere. I found the manuscript interesting and with the potential to be useful for the community studying airborne DNA. There is indeed the need to improve the way airborne DNA is sampled and the development of new sampling systems is needed. I do however have comments that I hope you will take into account.
I think the discussion section could be improved by making it more concise and focusing more on the evidence you provide and less about speculations. For example, you provide so many details about atmospheric conditions and wind direction so I would have expected more in depth discussion about the origin of the detected DNA (native species – or known to be found in the area vs. species only found outside the study area). I could not find information about which of the detected taxa are known to be found in the area and which are not. This would help in the discussion about how good the method used is to detect local taxa. It would also be interesting to discuss which height and/or weather conditions are the best ones for monitoring airborne DNA of local origin.
In addition, as the main focus of this manuscript is the novel hardware I suggest the authors to also add a section where you discuss the taxa that were not detected using your method (e.g. wildlife). This type of discussions is important when showcasing a new method.
Finally, you conclude that your results highlight the need for improved marker choices and reference databases. However this was never discussed in the text.

As a note: you collected paired samples in each flight. It would be of great interest to know differences in the taxa detected between the two filters, which would provide more evidence of the potential of this new hardware. In the methods you write that you combined the DNA extracts from both filters into one, but if you still have data from each filter I would suggest to use it too.

Additional comments

>Line 47 – Redundant to say “biological taxa”
>Line 49 – Although ice is water, I suggest to add “glacier” as the authors cite the work of Varotto et al 2021. In addition, none of the references used are about sediment, precipitation or air. I therefore recommend the authors to add appropriate references.
>Materials and method section could be made more concise. There is a lot of repetition of ideas, e.g description of the material used in the probe in line 175, again in 216. The discussion section could also be more concise.
>Line 294- Please clarify in which year this study was done.
>Line 368- To make it easier for the reader to know where to find this information, please reference to your File S1 and the table. Also, please provide all the information about in which negative controls these species were found as this is not clear.
>In the Supplementary file it is not possible to know which taxa were found in which negative controls, as the data is lumped into Flights.
>As the File S1 has very important information, I suggest to label each table with adequate titles that explain in a more complete form what is shown in them and with this, the authors could reference to the specific table within the File S1 in the main document. In addition, it is not clear why the taxonomic identification of the vertebrate sequences have several species and even genera assign to the same sequence. Better labeling could also be done for the Data_S1 file given that there are many tables which should be referenced directly in the main manuscript.
In addition, it would be very helpful for the reader to add an extra column next to the OTUs detected in the Heatmaps in File S1, indicating if the taxon is found in the study area. For example, Quercus rotundifolia is native to the Mediterranean region, Qercus robur is native to Europe, and Agrostis pallens from the West coast of the United States. This is important as in line 766 it is stated that you “found strong connectivity and mixing of local and regional surface emissions with the atmosphere…”
>Data_S1- Many of the labels in the figures are too small and it is not possible to read them.
The result section is too long and it could be written more concise. In many instances the results are being discussed (e.g. line 702-709, 714-753).
>Lines 702-705. Cnidaria, beetles and dragonflies were detected in the COI sequences but this information is not shown in the File S1 or in any file.
>Line 723- Please be more specific and provide information about what was the low relative numbers as this information is not provided in any of the Supplementary files.
>Line 759-760. You state that their findings indicate that their hardware captures genetic material representative of the diverse air environment surveyed. I consider this an overstatement, especially for vertebrates, as except for human DNA, you only detect Bos spp. and Gallus spp, even though their study area presents non-agricultural land.
>Line 775-778. It is true that the authors detected vertebrate DNA at high altitudes, but I suggest to tone down this sentence and make it clear that the only two non-human species detected (cow and chicken) are species found in high quantities in the area.
>File Data_S2 – There are no headers for the tables so it is not possible to understand the context and content of the tables
>Please remember that taxonomic families (eg. Pinaceae, Fagaceae, Bovidae) should not be written in italics, but this should only be done for the genus and species level.
>Line 922 and Line 899- I do not agree with the statement “it is remarkable how large-scale the dispersal of DNA is”, as the authors do not show evidence of the exact origin of the detected species. It would help if the authors marked in their maps exactly where the probable origin of the DNA is (e.g. poultry production area, waste treatment plants and so on).

Reviewer 2 ·

Basic reporting

The article is well written with appropriate literature referenced and appropriate structure. It is interesting to read, however is long and this may limit its readership to those very specifically interested in using the technology.

The introduction includes considerable background information and context for the study. It is well written. It could be argued that aerobiology has been established on fungal and palynological fields just as much as bacterial aerosols (see lines 59-60) (if not more when fungal crop pathology is considered). For completeness it would be worth referencing some of the extensive literature base on pollen aeroallergens and crop pathogen monitoring in the introduction.

Experimental design

The article presents a timely development in air sampling which will be of interest to several groups worldwide. Methodology is rigorous and well quality controlled. There are limitations to the technology (I understand a maximum of two samples per flight at present) which reduces its immediate appeal to other researchers to adopt, but could be a useful tool for aerobiology with future development.

The exclusion of all species present in the negative controls rather than setting a background threshold would be contentious if the focus of the study was taxa identification. However, the authors justify their choice because the purpose of this study is instrument validation. I agree this is plausible here (but personally would prefer the threshold method for any future taxonomic studies). All methodology is appropriate and reproducible from the detail provided.

The necessity of the Sanger sequencing (lines 436-437) is unclear. As is the purpose of the PCR (lines 413-428). Furthermore these do not feature in the results section except for Sanger results in the negative control (line 595).

Validity of the findings

Results are extremely detailed but easy to follow. The authors may want to consider condensing in parts. Line 665 relating to air parcel transport should be explained/ justified more. Also the ability of Festuca to hybridise is highlighted on line 699, but there needs to be more context for limitations of taxonomic resolution using ITS2 etc. The discussion is also well written and covers the key findings. Lines899-902 suggest the most taxa came from specific locations in the sampling. However there is no definitive evidence for this from the study and emission sources are not a major consideration of the study design, so I would suggest rewording or removing this. Although well written, it is lengthy and also may require condensing as the article overall is considerably extensive.

---

## Round 0.2 · accepted · Accept

All of the comments from reviewers appear to have been addressed, congratulations on having this accepted!